# Reinforcing the North Atlantic backbone: revision and extension of the composite splice at ODP Site 982

Anna Joy Drury[1], Thomas Westerhold[1], David Hodell[2], Ursula Röhl[1]

[1]MARUM - Center for Marine Environmental Sciences, University of Bremen, Germany
5   [2]Department of Earth Sciences, University of Cambridge, Cambridge CB2 3EQ, UK

*Correspondence to*: Anna Joy Drury (ajdrury@marum.de)

**Abstract.** Ocean Drilling Programme (ODP) Site 982 represents a key location for understanding the evolution of climate in the North Atlantic over the past 12 Ma. However, concerns exist about the validity and robustness of the underlying stratigraphy and astrochronology, which currently limits the adequacy of this site for high-resolution climate studies. To resolve this uncertainty, we verify and extend the early Pliocene to late Miocene shipboard composite splice at Site 982 using high-resolution XRF core scanning data and establish a robust high-resolution benthic foraminiferal stable isotope stratigraphy and astrochronology between 4.5 and 8.0 Ma. Splice revisions and verifications resulted in ~11 m of gaps in the original Site 982 isotope stratigraphy, which were filled with 263 new isotope analyses. This new stratigraphy reveals previously unseen benthic $\delta^{18}O$ excursions, particularly prior to 6.65 Ma. The benthic $\delta^{18}O$ record displays distinct, asymmetric cycles between 7.7 and 6.65 Ma, confirming that high-latitude climate is a prevalent forcing during this interval. An intensification of the 41-kyr beat in both the benthic $\delta^{13}C$ and $\delta^{18}O$ is also observed ~6.4 Ma, marking a strengthening in the cryosphere-carbon cycle coupling. A large ~0.7‰ double excursion is revealed ~6.4-6.3 Ma, which also marks the onset an interval of average higher $\delta^{18}O$ and large precession and obliquity-dominated $\delta^{18}O$ excursions between 6.4-5.4 Ma, coincident with the culmination of the late Miocene cooling. The two largest benthic $\delta^{18}O$ excursions ~6.4-6.3 Ma and TG20/22 coincide with the coolest alkenone-derived SST estimates from Site 982, suggesting a strong connection between the late Miocene global cooling and deep-sea cooling and dynamic ice sheet expansion. The splice revisions and revised astrochronology resolve key stratigraphic issues that have hampered correlation between Site 982, the equatorial Atlantic and the Mediterranean. Comparisons of the revised Site 982 stratigraphy to high-resolution astronomically tuned benthic $\delta^{18}O$ stratigraphies from ODP Site 926 (equatorial Atlantic) and Ain el Beida (north western Morocco) show that prior inconsistencies in short-term excursions are now resolved. The identification of key new cycles at Site 982 further highlights the requirement for the current scheme for late Miocene marine isotope stages to be redefined. Our new integrated deep-sea benthic stable isotope stratigraphy and astrochronology from Site 982 will facilitate future high-resolution late Miocene to early Pliocene climate research.

# 1 Introduction

The North Atlantic remains a crucial location for understanding past climate change. It has been a prime region of deep water formation since at least the middle Miocene (Woodruff and Savin, 1989) and a key location for correlating between the progressively enclosed Mediterranean basin and the major oceanic basins (Andersson and Jansen, 2003; Hodell et al., 2001). North Atlantic Ocean Drilling Programme (ODP) Site 982, situated on the Rockall Plateau, was recovered to document the evolution of intermediate water circulation in the North Atlantic Basin (Shipboard Scientific Party Leg 162, 1996). Palaeoclimate records generated at this location have been fundamental to improving our understanding of climate for the past 12 Ma (Bolton et al., 2011; Herbert et al., 2016; Hodell et al., 2001; Hodell and Venz-Curtis, 2006; Khélifi et al., 2014; Lawrence et al., 2009; Lisiecki and Raymo, 2005; Venz et al., 1999; Venz and Hodell, 2002). In particular, the late Miocene benthic stable isotope stratigraphy and astrochronology (Hodell et al., 2001) has been used as the basic stratigraphy and/or age model for a wide range of studies (Bickert et al., 2004; Drury et al., 2016; Herbert et al., 2016; Hodell and Venz-Curtis, 2006; Kontakiotis et al., 2016; De Vleeschouwer et al., 2017). However, there is an ongoing debate about the presence of potential stratigraphic issues in the Pliocene successions of Site 982 (Khelifi et al., 2012; Lawrence et al., 2013) and concerns exist that individual cycles may be missing from the late Miocene isotope stratigraphy (Bickert et al., 2004; van der Laan et al., 2005, 2012); however, these concerns have not yet been resolved.

Site 982 is characterised by high carbonate content, which is above 90% for most of the Pliocene and Miocene (Hodell et al., 2001; Shipboard Scientific Party Leg 162, 1996). The standard shipboard physical properties measurements, such as spectral reflectance, gamma ray attenuation (GRA) bulk density, magnetic susceptibility and natural gamma radiation are difficult to use for stratigraphic correlation in such high carbonate settings. At Site 982, the early Pliocene to late Miocene composite splice was constructed using solely GRA bulk density and spectral reflectance in the 650-700-nm band; however, the shipboard composite record for this interval has never been independently verified. The stratigraphic discrepancies between Site 982 and other deep-sea and Mediterranean sites could originate from errors in the shipboard composite record and verification of the shipboard composite splice is essential to resolve these issues. Non-destructive X-ray Fluorescence (XRF) core scanning is an ideal technique for recovering the high quality and high-resolution data that are required for composite splice verification.

Here, we aim to verify and establish a robust high-resolution stable isotope stratigraphy and astrochronology at Site 982 that can continue as a reference section for North Atlantic climate change. Firstly, we use XRF core scanning to successfully test and extend the early Pliocene to late Miocene shipboard splice at Site 982. Following splice revision, we generate new benthic stable isotope data to fill gaps in the existing stable isotope stratigraphy at Site 982. Using the new complete benthic $\delta^{18}O$ and $\delta^{13}C$ stratigraphy, we generate a new astrochronology for Site 982 between 4.5 and 8.0 Ma. Finally, comparing the new stable isotope stratigraphy to stable isotope stratigraphies from ODP Site 926 (equatorial Atlantic) and Ain El Beida (northwest Morocco) allows us to assess whether previous stratigraphic issues are resolved. This study intends to reinforce Site 982, and the climate records generated there, as a backbone for palaeoceanographic research in the North Atlantic.

## 2 Materials and Methods

### 2.1 ODP Site 982

This study utilises early Pliocene to late Miocene sediments recovered from ODP Site 982 (Figure 1; 57°30.992'N,' 15°52.001'W'; 1134 m water depth) drilled on the Rockall Plateau in the North Atlantic during ODP Expedition 162 (Shipboard Scientific Party Leg 162, 1996). Site 982 was cored to reconstruct Neogene intermediate-water circulation in the North Atlantic. A continuous shipboard composite section was developed until approximately 248 metres composite depth (mcd) back to the late Miocene. The late Miocene through early Pliocene sediments predominantly consist of biogenic carbonate (nannofossil ooze), with minor amounts of clay and biogenic silica, interspersed with occasional ash layers (Shipboard Scientific Party Leg 162, 1996; Wallrabe-Adams and Werner, 1999). Carbonate content is high, averaging 90.8% in the late Miocene to early Pliocene sequences. A high-resolution, astronomically-tuned, benthic stable isotope record (temporal sampling ~2.5 kyr) from this location was published between 4.6 and 8.9 Ma (Hodell et al., 2001). Considering the site's location in the North Atlantic, this isotope record is important for comparing to isotopic records from the Mediterranean basin in order to provide regional records from the enclosed Mediterranean basin with the global context preserved more strongly at deep-sea locations like Site 982 (Hodell et al., 2001; van der Laan et al., 2005). We assess the new benthic foraminiferal stable isotope stratigraphy developed here by comparing it to similarly high-resolution, astronomically-tuned, benthic stable isotope stratigraphies from proximal locations in the equatorial Atlantic (ODP Site 926) and Mediterranean basin (Ain el Beida) (Figure 1). These records have additionally been instrumental in initially highlighting that stratigraphic issues with the shipboard splice may affect the original Site 982 isotope stratigraphy (Bickert et al., 2004; Drury et al., 2017; van der Laan et al., 2005).

### 2.2 X-Ray fluorescence core scanning

XRF Core Scanner data were collected every 2 cm down-core over a 1 cm$^2$ area with down-core slit size of 12 mm (Run 1: 50 kV, 1.0 mA, 15 s count time; Run 2: 10 kV, 0.2 mA, 15 s count time) directly at the split core surface of the ODP 982 archive halves with XRF Core Scanner II (AVAATECH Serial No. 2) at the MARUM - University of Bremen. The split core surface was covered with a 4 μm thick SPEXCerti Prep Ultralene1 foil to avoid contamination of the XRF measurement unit and desiccation of the sediment. The reported data were acquired by a Canberra X-PIPS Silicon Drift Detector (SDD; Model SXD 15C-150-500) with 150eV X-ray resolution, the Canberra Digital Spectrum Analyzer DAS 1000, and an Oxford Instruments 50W XTF5011 X-Ray tube with rhodium (Rh) target material. Raw X-ray spectra were processed using Iterative Least square software (WIN AXIL) package from Canberra Eurisys. The data was collected between April 2014 and June 2016, during which time the X-Ray tube was replaced. To account for differences in measurement intensity, a number of sections were re-measured and a linear regression between the two datasets was used to calibrate the data collected in 2014 to the data collected in 2015-2016 (see Supplementary Information and Supplementary Figure 1). The shipboard composite

depth section (in mcd) was scanned between 120.13 to 271.58 mcd (Shipboard Scientific Party Leg 162, 1996). All XRF core scanning Si and Zr intensity data are reported in Supplementary Table 1.

## 2.3 CODD image processing and core composite image generation

To assist with splice verification and data interpretation, core composite images for Site 982 were compiled from core table photos using Code for Ocean Drilling Data (CODD – www.codd-home.net; Wilkens et al., 2017). For each hole, every section from the individual core table images (downloaded from the JANUS database – http://www-odp.tamu.edu/database/) were cut, compiled and scaled to mcd using the Includes_Core_Table_Photos functions within CODD. A lighting correction was also applied to account for the single point lighting source used in core table photos (see Wilkens et al., 2017 and the CODD user guide for further information).

## 2.4 Stable isotope analyses

To fill the observed gaps in the original Hodell et al. (2001) stable isotope stratigraphy and re-establish a robust stratigraphy for the North Atlantic, an additional 263 samples were taken for isotope analysis between 200 and 280 revised m composite depth (rmcd). These samples were selected to fill the 11 m of missing sections in the original stable isotope stratigraphy, as well as overlapping with the original data to ensure the two datasets could be integrated. Each measurement consisted of 1-6 translucent specimens of *Cibicidoides (C.) mundulus* or *C. wuellerstorfi* (250-500 μm), which were analysed for carbon ($\delta^{13}$C) and oxygen ($\delta^{18}$O) isotopes using a Kiel I carbonate preparation device attached to a Finnigan MAT 251 at MARUM (University of Bremen, Germany), with an analytical precision of 0.03‰ for $\delta^{13}$C and 0.06‰ for $\delta^{18}$O. All results are reported against Vienna Peedee Belemnite (VPDB) using the standard δ notation (per mille ‰), determined using calibrated in-house standards and NBS-19. All data were corrected to equilibrium using the species-specific constants listed in Supplementary Table 2 (Shackleton et al., 1984, 1995). No offsets were found between the new $\delta^{13}$C and $\delta^{18}$O measurements and the existing records at Sites 982. In graphic representation, only $\delta^{13}$C data obtained on *C. wuellerstorfi* was used because Hodell et al. (2001) found that $\delta^{13}$C values from *C. mundulus* become more negative than paired $\delta^{13}$C values of *C. wuellerstorfi* during intervals of lower $\delta^{13}$C. All raw and corrected isotope data are provided in Supplementary Table 3.

## 3 Results

### 3.1 XRF core scanning data and splice revision

The natural logarithm of the Si intensity data (ln(Si)) shows significant variability on both ~0.8 and ~1.6 m scale, which coincides well with darker sedimentary layers, as seen on the composite core photos, whilst the low ln(Si) values coincide with the lighter, whiter layers (Figure 2.A; Supplementary Figure 2). Among Holes 982A, 982B and 982C, the cycles visible

in the ln(Si) and core photos are remarkably consistent (Figure 2.A), which shows these are ideal for verifying the shipboard composite splice.

The XRF ln(Si) records and core photos reveal mismatches, overlaps and gaps in the shipboard composite section (Figure 2.A and 3.A; Supplementary Figure 2 and 3). Following splice revision, the agreement of the composite core photos and ln(Si) is remarkable between the three holes (Figure 2.B. and 3.A; Supplementary Figure 2 and 3). The shipboard composite splice is altered and verified between 120.13 to 271.58 revised metres composite depth (rmcd) and additionally extended to 280.23 rmcd (Supplementary Figure 2; Tables 1 and 2). The splice revisions resulted in ~11 m of gaps in the published isotope data (Hodell et al., 2001), predominantly in the late Miocene (Figure 3.B; Supplementary Figure 3). Some individual gaps and overlaps are as large as ~2-3 meters, which is equivalent to ~40-50 kyr on the original Hodell et al. (2001) age model (Figure 3.B; Supplementary Figure 3). The revised offset and splice tables are presented in Tables 1 and 2, and are digitally available on PANGAEA and in Supplementary Tables 4 and 5.

On the revised composite splice, the ln(Si) data shows strong ~1.6 m cyclicity throughout the record, which is particularly strong between 180 and 240 rmcd (Figure 4.C; Figure 5.A). Additionally, distinct ~0.8 m cyclicity is visible between 140 and 220 rmcd (Figure 4.C). In addition to strong ~0.8 and ~1.6 m cyclicity, some ln(Si) peaks coincide with high Zr peaks, and darker sedimentary layers, reflecting the presence of ash layers at Site 982 (pink bars - Figure 4.C, D and E)

**3.2 Revised Site 982 stable isotope stratigraphy**

Using the new isotope data produced here (Figure 3.B), the resulting Site 982 benthic $\delta^{18}O$ stratigraphy display little long-term variation (0.2-0.3 ‰), which is characteristic of this late Miocene interval (Figure 4.B). Throughout the record ~1.6 m cycles of ~0.5-0.6‰ are observed, in particular between 270 and 235 rmcd and again between 215 to 160 rmcd (Figure 4.B; Figure 5.A). Between 210 and 160 rmcd ~0.8 m cycles are particularly strong in the 982 benthic $\delta^{18}O$ record (Figure 4.B; Figure 5.A). The complete Site 982 $\delta^{13}C$ stratigraphy displays greater long-term variability, most notably with the negative –1‰ shift between 265 and 220 rmcd (Figure 4.A). The $\delta^{13}C$ record displays strong ~1.6 m cycles of ~0.5‰, although longer 3.8-5.0 m cycles also appear in the upper section above ~170 rmcd (Figure 4.A; Figure 5.B and C). The $\delta^{18}O$ and $\delta^{13}C$ records generally display an antiphase relationship on the scale of ~1.6 m cycles (Figure 4.A and B). A comparison of the 982 stable isotope stratigraphy with the XRF Zr record shows that there is no effect of the ash layers on the isotope record (Figure 4).

**4 Astrochronology**

The original astrochronology presented in Hodell et al. (2001) was constructed by respectively tuning the benthic $\delta^{18}O$ record filtered at 41 kyr to obliquity and the GRA bulk density record filtered at 21 kyr to summer insolation at 65°N. The obliquity and 65°N summer insolation tuning targets were taken from the La90(1,1) astronomical solution (Laskar et al.,

1993). Although the differences between the La90(1,1) and La2004(1,1) solutions are not large in the last 10 Ma, small offsets are noticeable (Laskar et al., 2004). The original Hodell et al. (2001) astrochronology was updated when De Vleeschouwer et al. (2017) incorporated the Site 982 benthic $\delta^{18}O$ stratigraphy as the 5.174-8.560 Ma interval of the Neogene megasplice and retuned the original stratigraphy to the La2004 obliquity solution as part of this process. However, as our splice revisions revealed duplicated cycles, as well as 11 m of gaps in the original spliced composite section, the new stable isotope stratigraphy presented here requires a new astrochronology.

Constructing an astrochronology at Site 982 was an iterative process. To establish whether the strong cyclicity observed in the benthic $\delta^{18}O$ and $\delta^{13}C$ records in the depth domain related to orbital forcing, first-order age control was provided by a 5th order polynomial fit through 13 shipboard nannofossil and planktonic foraminiferal datums (Supplementary Table 6). The initial age control shows that the ~0.8 m cycles observed in the ln(Si) and benthic $\delta^{18}O$ record are consistent with precession forcing and the more dominant ~1.6 m cycles observed throughout the $\delta^{18}O$ and $\delta^{13}C$ records are consistent with obliquity forcing (Figure 5.A). Additionally, the 3.8-5.0 m cycles observed in the upper parts of the $\delta^{13}C$ record correspond to eccentricity forcing (Figure 5.B and C). The presence of these orbitally-forced cyclicities supports that the $\delta^{18}O$ and $\delta^{13}C$ records are suitable for astronomical tuning.

To construct the astrochronology for Site 982, we correlated the benthic $\delta^{18}O$ series on depth to computed variations in the Earth's orbit on age, since the benthic $\delta^{18}O$ in particular showed strong variability at precession and obliquity timescales. Due to the presence of both strong obliquity and precession driven cycles, we chose a tuning target composed of equally weighted eccentricity, tilt and Northern Hemisphere precession (E+T-P, from Laskar et al., 2004). No phase shift was applied to account for possible lags between $\delta^{18}O$ and insolation forcing due to the response time of ice sheets, as this response time is unknown for the Miocene (Holbourn et al., 2007).

Initially, a minimal tuning strategy was applied (see also Holbourn et al., 2007; Drury et al., 2017), where distinctive $\delta^{18}O$ minima were visually correlated to E+T-P maxima using 20 tie points (purple lines - Figure 6; Supplementary Figure 4 and Table 7), facilitated by the tuning functions available in CODD (Wilkens et al., 2017). The shipboard biostratigraphic datums (Supplementary Table 6) were used to initially guide the minimal tuning process, but were not included as fixed tie points. Although the shipboard datum ages were updated to the astronomically tuned Neogene timescale (GTS2012; Hilgen et al., 2012), the Site 982 minimal tuning is negligibly influenced by the GTS2012 calibration, as the shipboard datums have large depth errors (±0.25-2.5m; Supplementary Table 6) and were not used as definitive tie points. The exact minimal tuning tie points were selected to align $\delta^{18}O$ minima-E+T-P maxima as best as possible between individual ties. Additionally, consecutive ties were ideally placed at least ~100 kyr apart (actual range 90-377 kyr), as recommended by Zeeden et al. (2015) to avoid the introduction of frequency modulation through the astronomical tuning process. Finally, a fine-tuning strategy was applied (see also Drury et al., 2017), where an additional 41 tie points were added to correlate the remaining distinctive $\delta^{18}O$ minima to E+T-P maxima (grey lines - Figure 6; Supplementary Table 7). This fine-tuning age model provides higher-resolution age control and removes remaining $\delta^{18}O$ minima-E+T-P maxima misalignments in between the initial minimal tuning tie points (Figure 6).

## 5 Discussion

### 5.1 New excursions and deep-sea cooling: insights from benthic isotope stratigraphy at Site 982

Surprisingly, following splice revision, the new, complete, high-resolution (~2.5 kyr) benthic $\delta^{18}O$ and $\delta^{13}C$ records from Site 982 show a number of large excursions that have not been observed previously (Figure 7.B). Spectral analyses of both records confirm that obliquity is the dominant forcing across the entire interval, although precession is additionally present (Figure 5, Figure 7.D and E). Many of the new excursions occur between 7.7 and 6.65 Ma, when the benthic $\delta^{18}O$ record shows a strong response to obliquity forcing (Figure 7.C and E), confirmed by the strength of the obliquity spectral peak (Figure 5.D). The timing and shape of the $\delta^{18}O$ cycles in particular coincides with the distinctive saw-tooth obliquity driven cycles observed between 7.7-6.9 Ma at eastern equatorial Pacific IODP Site U1337 (Drury et al., 2017). The asymmetry of the $\delta^{18}O$ cycles seen at Site U1337, coincident with antiphase obliquity cycles in $\delta^{18}O$ and $\delta^{13}C$, were associated with high-latitude climate forcing, although the asymmetry could partly also result from the interference pattern between strong obliquity forcing and the influence of minor precession (Drury et al., 2017). The distinct cycles at Site 982 between 7.7 and 6.65 Ma also display asymmetry and coincide with a strong antiphase relationship between the Site 982 $\delta^{18}O$ and $\delta^{13}C$ records (Figure 7.B-F), supporting that high-latitude climate forcing is strong during this interval of the late Miocene. However, the precessional influence observed at Site U1337 is not as developed in the Site 982 $\delta^{18}O$ records during the 7.7-6.9 Ma interval (Figure 5.D, Figure 7.C and E). In addition to the dominance of obliquity over precession forcing (Figure 5.D), the obliquity-driven $\delta^{18}O$ cycles at Site 982 continue until 6.65 Ma, indicating an extended influence of high-latitude processes until 6.65 Ma. Between ~7.6 and 6.7 Ma, the Site 982 benthic $\delta^{13}C$ record is characterised by the Late Miocene Carbon Isotope Shift (LMCIS) (Figure 7.B): a globally expressed negative ~1‰ shift that is thought to reflect a global change in the $\delta^{13}C$ of the oceanic reservoir (Haq et al., 1980; Hodell and Venz-Curtis, 2006; Keigwin, 1979). When compared to the high-resolution $\delta^{13}C$ stratigraphy from ODP Site 926 (equatorial Atlantic), this confirms that the timing of the onset and termination of the LMCIS was globally synchronous within a few kyr (Figure 8.B), as shown between Site 926 and Site U1337 (equatorial Pacific) (Drury et al., 2017).

The Site 982 $\delta^{18}O$ and $\delta^{13}C$ records shows minimal variability between 6.65 and 6.5 Ma during a long-term obliquity node; however, obliquity forcing strongly manifests itself again ~6.5 Ma in the $\delta^{18}O$ record. This is shortly followed by a remarkably large double excursion in the Site 982 benthic $\delta^{18}O$ spanning two obliquity cycles between ~6.3-6.4 Ma (~210-214 rmcd), which marks a rapid increase of ~0.7‰ (black arrow - Figure 7.C; Figure 4.B). This excursion coincides with the first occurrence of strong obliquity forcing in the $\delta^{13}C$ record as well as the $\delta^{18}O$ record (Figure 5.E and F, Figure 7.B-F), confirming a strengthening of the antiphase cryosphere-carbon cycle coupling and an intensification of the 41-kyr beat indicated at Site U1337 after 6.4 Ma (Drury et al., 2017). The large $\delta^{18}O$ excursion marks the onset of a strong response to both obliquity and northern hemisphere precession in particularly the Site 982 $\delta^{18}O$ record between 6.4 and 5.4 Ma, with strong obliquity forcing continuing until at least 4.6 Ma (Figure 7.D). The interplay between precession and obliquity forcing

on the $\delta^{18}O$ record between 6.4 and 5.4 Ma could account for the presence of well-expressed power in the 24, 22 and 19-kyr bandwidth seen in the MTM power spectra (Figure 5.D)

Between 6.4 and 5.4 Ma, the $\delta^{18}O$ stratigraphy is characterised by average higher $\delta^{18}O$, and the greatest variability seen throughout the record. This interval at Site 982 was associated with the latest Miocene glaciation by Hodell et al. (2001). In addition to the new excursion ~6.4-6.3 Ma, marine isotope stages TG12/14 (~5.5-5.6 Ma) and TG20/22 (~5.7-5.8 Ma) are also clearly expressed at Site 982 (Figure 7.B). This high-$\delta^{18}O$, high-variability interval coincides with end of the late Miocene cooling interval associated with declining atmospheric $CO_2$ concentrations identified by Herbert et al. (2016). Notably, the two biggest drops in alkenone-derived sea surface temperatures (SSTs) from Site 982, when surface waters were only ~12-13°C, coincide with the two largest excursions in benthic $\delta^{18}O$, namely the new excursion revealed ~6.4-6.3 Ma and TG20/22. The previous absence of the 6.4-6.3 Ma excursion in benthic $\delta^{18}O$ record of Site 982 is attributed to missing section in the shipboard splice, which accounts for why the late Miocene global cooling observed by Herbert et al. (2016) did not seem to be associated with any response in deep-sea temperatures or global ice volume expansion. However, the synchronicity between the 6.4 Ma intensification of the 41-kyr beat in both the $\delta^{13}C$ and $\delta^{18}O$ records, the new 6.4-6.3 Ma $\delta^{18}O$ excursion, the greatest cooling in the alkenone SST records (Figure 7), suggests that the late Miocene cooling seen in the high-latitudes lead to a combination of deep-sea cooling, dynamic ice sheet expansion and a stronger coupling between the cryosphere and the carbon cycle.

## 5.2 Integrating the Mediterranean and North Atlantic

Site 982 is a key reference section for palaeoceanographic and palaeoclimate study, as it is the only deep-sea location in the North Atlantic with high-resolution benthic oxygen isotope stratigraphies for the late Miocene to early Pliocene. As such, it is frequently used as a North Atlantic end member to investigate past deep-sea circulation (Bickert et al., 2004; Drury et al., 2016; Hodell et al., 2001; Hodell and Venz-Curtis, 2006; Khélifi et al., 2014). Additionally it reflects a crucial link to tie regional records of the enclosed Mediterranean basin, such as those from Ain el Beida and Salé Briqueterie, to the global signal preserved more strongly in deep-sea sediments (Hodell et al., 1994, 2001; Kontakiotis et al., 2016; van der Laan et al., 2005, 2012). However, although the long-term trends at Site 982 remained useful for investigating long-term, million-year changes in deep-water circulation patterns, on inspection at higher-resolution, misalignments between distinct excursions become apparent (Bickert et al., 2004; van der Laan et al., 2005). Bickert et al. (2004) constructed their ODP Site 999 age model by correlating their isotope data to the isotope stratigraphy from Site 982; however, to account for obvious mismatches between Site 999 and nearby ODP Site 926, they had to adjust the original Site 982 astrochronology to account for some inconsistencies with respect to obliquity. The isotope stratigraphy at 926 has recently been extended back to ~8.0 Ma, following splice verification and revision and validation of the original Shackleton and Hall (1997) astrochronology (Drury et al., 2017; Wilkens et al., 2017; Zeeden et al., 2013). Although there is overall agreement in long-term trends and the onset of the LMCIS, inconsistencies between distinctive short-term events in the original Site 982 shipboard stratigraphy

and the updated Site 926 stratigraphy are evident (Figure 8.A). Disagreement between the $\delta^{13}C$ records from Sites 926 and 982 is particularly pronounced before 7.3 Ma, between 6.3-6.9 Ma (the termination of the LMCIS) and between 5.9-5.1 Ma (Figure 8.A). Following splice revision and the new astronomical calibration, the new Site 982 stratigraphy shows far better agreement throughout, both in terms of the long-term trends, the onset and termination of the LMCIS, and especially with

respect to individual excursions (Figure 8.B). This considerable improvement between these two Atlantic deep-sea benthic stratigraphies strongly supports the revised astrochronologies at both sites.

Comparisons between late Miocene Mediterranean records and Site 982 also raised concerns about the validity of the original astrochronology and splice. A high-resolution comparison between stable isotope data from Ain el Beida (AEB; northwest Morocco) and Site 982 indicated that there were both mismatches in the number of cycles present at both

locations, as well as inconsistencies between the age models with respect to the age of clearly identical cycles (Figure 8.C; van der Laan et al., 2005). Again, following the splice revisions and new orbital tuning, the agreement between these two records is radically improved (Figure 8.D). The misalignment in age between TG12/14 (~5.5-5.6 Ma) at Site 982 highlighted by van der Laan et al. (2005) (Figure 8.C) has been resolved when using the new benthic stratigraphy (Figure 8.D). Furthermore, by the addition of an extra cycle through the revision of the splice at Site 982, the agreement between AEB and

Site 982 is improved between TG34 and C3An.$\delta^{18}$O14/16 (**14/16 in Figure 8.C and D). Older than stage C3An.$\delta^{18}$O16, between 6.5 and 6.3 Ma, correlation between Site 982 and AEB was difficult based on the original stratigraphies, with the trends diverging and poor agreement between individual excursions (Figure 8.C; van der Laan et al., 2005). With the identification of the large new double excursion in benthic $\delta^{18}$O at Site 982 between ~6.3-6.4 Ma (Figure 7.B), the benthic $\delta^{18}$O records from AEB and Site 982 now show remarkable agreement in this interval (Figure 8.D). This is further indication

that this event marks a major global event, preceding the coolest temperatures as identified by Herbert et al. (2016).

The new stratigraphy does not resolve all misalignments between Site 982 and AEB. Between TG12/14 and TG20/22, van der Laan et al. (2005) identified an additional cycle at Site 982 that was not visible in AEB (Figure 8.C). In the new stratigraphy, the agreement between Site 982 and AEB is still not perfect, with Site 982 displaying 1-2 cycles more than AEB (Figure 8.D). Low sedimentation rates at AEB in this interval (van der Laan et al., 2005), as well as lower sampling

resolution, could indicate cycles may be missing at this location. Splice uncertainty at Site 982 is unlikely to be the cause of this discrepancy, as the XRF core scanning splice verifications show that the splice is robust in this interval. Also the interval where Site 982 shows additional cycles compared to AEB occurs within a single core. Although sedimentary disturbances cannot be excluded and are difficult to identify in low-contrast deep-sea pelagic sediments with high carbonate content, the stable isotope and XRF core scanning datasets support that the sedimentary succession at Site 982 in this interval is

undisturbed.

Our study has resolved many of the issues surrounding the stratigraphy and astrochronology at Site 982. New cycles were identified, which correlate well with isotope records from the Mediterranean and equatorial Atlantic. Past attempts have been made to extend the Marine Isotope Stage (MIS) identification scheme first developed by Shackleton et al. (1995) into the late Miocene (Drury et al., 2016; van der Laan et al., 2005). The identification of the new cycles at Site 982 highlights

the need for a redefinition of the current MIS scheme for this interval. Considering the number of high-resolution, robust benthic foraminiferal stratigraphies that now exist for the latest Miocene (Bickert et al., 2004; Drury et al., 2016, 2017; Vidal et al., 2002; Westerhold et al., 2005), a redefinition of late Miocene MIS should soon be attempted, following the global approach applied for the Pliocene by Lisiecki and Raymo (2005).

## 6 Conclusions

Here, we present a new high-resolution benthic stable isotope stratigraphy and astrochronology for ODP Site 982. Splice revisions using XRF core scanning data resulted in ~11 m of gaps in the original Site 982 isotope stratigraphy, with some gaps/overlaps as large as ~2-3 meters, equivalent to ~40-50 kyr on the original age model. Our complete stratigraphy

reveals previously unseen benthic $\delta^{18}O$ excursions, particularly in the late Miocene section. The benthic $\delta^{18}O$ record displays distinct, asymmetric cycles between 7.7 and 6.65 Ma, supporting evidence from the Pacific that high-latitude climate forcing is strong during this interval. Additionally, the intensification of the 41-kyr beat in both the benthic Site 982 $\delta^{18}O$ and $\delta^{13}C$ records confirm that a strengthening of the cryosphere-carbon cycle coupling occurred ~6.4 Ma.

Around 6.4-6.3 Ma, the splice revisions also reveal a significant ~0.7‰ double excursion in the Site 982 $\delta^{18}O$

stratigraphy covering two obliquity cycles. This excursion marks the onset of the 6.4-5.4 Ma interval characterised by average higher $\delta^{18}O$ and greater short-term variability dominated by obliquity and minor precessional influence. This long-term $\delta^{18}O$ maximum and associated high short-term variability strongly coincides with the maximum of late Miocene cooling associated with declining $CO_2$ (Herbert et al., 2016). In particular, the two largest benthic $\delta^{18}O$ excursions ~6.4-6.3 Ma and TG20/22 coincide with the coolest alkenone-derived SST estimates from Site 982. This suggests that the late

Miocene global cooling coincided with deep-sea cooling and dynamic ice sheet expansion. At 6.4 Ma, the intensification of the 41-kyr beat, the onset of high benthic $\delta^{18}O$ variability and greatest alkenone SST cooling at Site 982, strongly supports that the hypothesised decline in atmospheric $CO_2$ concentrations and late Miocene cooling led to deep-sea cooling, dynamic ice sheet expansion and a stronger coupling between the cryosphere and the carbon cycle.

The revised splice and astrochronology resolve key stratigraphic issues that have hampered correlation between Site

982 and other locations. Comparisons of the revised Site 982 $\delta^{18}O$ stratigraphy to benthic $\delta^{18}O$ stratigraphies from ODP Site 926 and Ain el Beida show that prior inconsistencies in short-term excursions are now resolved. Individual excursions show remarkable agreement between the north and equatorial Atlantic Ocean, as well as the north Atlantic and the Mediterranean basin, thereby strongly supporting the revised astrochronology. The identification of new cycles at Site 982 also highlights the need for a redefinition and extension of the current late Miocene MIS scheme. Finally, our new deep-sea stratigraphy

will enable Site 982 to continue as a key North Atlantic sequence for high-resolution study of the palaeoceanography of the late Miocene to early Pliocene.

## Data Availability and Code

All data, including a Site 982 CODD experiment and all composite core images, are available on the open access PANGAEA database at https://doi.org/10.1594/PANGAEA.884300. The supplementary data tables are also available from the supplementary information at *Climate of the Past*. CODD functions for IGOR Pro[TM] used in this paper (Wilkens et al.,
2017) are available for download from https://www.codd-home.net and the shipboard physical property data from ODP Site 982 is available online from the JANUS database (http://www-odp.tamu.edu/database/).

## Author Contributions

The project was designed by AJD, TW and DH. AJD, TW and UR contributed to the XRF dataset, AJD and TW generated the stable isotope record and AJD cut the core images from the core box photos. All authors contributed to data collection, quality control and analysis. All authors were involved in scientific discussions. AJD and TW wrote the manuscript with additional contributions from all authors.

## Acknowledgements

This research used data acquired at the XRF Core Scanner Lab at the MARUM – Center for Marine Environmental Sciences, University of Bremen, Germany. We thank Vera Lukies (MARUM) for assistance with XRF core scanning, Henning Kuhnert and his team (MARUM) for stable isotope analyses, Alex Wülbers and Walter Hale (IODP Bremen Core Repository) for core handling, and Barbara Donner (MARUM) for providing foraminiferal expertise. We additionally thank two anonymous reviewers and Pierre Francus for their constructive reviews, which helped to improve this manuscript. This
research used samples and data provided by the Ocean Drilling Program (ODP), sponsored by the US National Science Foundation (NSF) and participating countries. The Deutsche Forschungsgemeinschaft (DFG) provided financial support for this research (We5479/1).

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

| Table 1: Revised and Shipboard offsets for ODP Site 982 (Holes A-D) | | | | | | |
|---|---|---|---|---|---|---|
| *Key* | **BOLD = changes compared to shipboard splice** | | | *Italics = shipboard splice shifted to accommodate new offsets higher in the section.* | | |

| Core | Depth | Shipboard | Revised | Change in | shipboard composite depth | Revised composite depth |
|---|---|---|---|---|---|---|
|  | *(mbsf [m CSF-* | *(m)* | *(m)* | *(m)* | *(mcd [m CCSF-A])* | *(rmcd [m rCCSF-A])* |
| **162-982A-** | | | | | | |
| 1H | 0.00 | 0.00 | 0.00 | 0.00 | 0.00 | 0.00 |
| 2H | 8.20 | 1.30 | 1.30 | 0.00 | 9.50 | 9.50 |
| 3H | 17.70 | 2.08 | 2.08 | 0.00 | 19.78 | 19.78 |
| 4H | 27.20 | 2.50 | 2.50 | 0.00 | 29.70 | 29.70 |
| 5H | 36.70 | 3.93 | 3.93 | 0.00 | 40.63 | 40.63 |
| 6H | 46.20 | 6.53 | 6.53 | 0.00 | 52.73 | 52.73 |
| 7H | 55.70 | 7.97 | 7.97 | 0.00 | 63.67 | 63.67 |
| 8H | 65.20 | 8.78 | 8.78 | 0.00 | 73.98 | 73.98 |
| 9H | 74.70 | 9.83 | 9.83 | 0.00 | 84.53 | 84.53 |
| 10H | 84.20 | 11.43 | 11.43 | 0.00 | 95.63 | 95.63 |
| 11H | 93.70 | 12.35 | 12.35 | 0.00 | 106.05 | 106.05 |
| 12H | 103.20 | 13.49 | 13.49 | 0.00 | 116.69 | 116.69 |
| ***13H*** | ***112.70*** | ***14.92*** | ***14.82*** | ***-0.10*** | ***127.62*** | ***127.52*** |
| *14H* | *122.20* | *15.82* | *15.72* | *-0.10* | *138.02* | *137.92* |
| ***15H*** | ***131.70*** | ***17.28*** | ***16.99*** | ***-0.29*** | ***148.98*** | ***148.69*** |
| ***16H*** | ***141.20*** | ***18.12*** | ***17.97*** | ***-0.15*** | ***159.32*** | ***159.17*** |
| *17H* | *150.70* | *19.00* | *18.85* | *-0.15* | *169.70* | *169.55* |
| *18H* | *160.20* | *19.50* | *19.35* | *-0.15* | *179.70* | *179.55* |
| ***19H*** | ***169.70*** | ***20.78*** | ***20.70*** | ***-0.08*** | ***190.48*** | ***190.40*** |
| ***20H*** | ***179.20*** | ***20.34*** | ***21.70*** | ***1.36*** | ***199.54*** | ***200.90*** |
| ***21H*** | ***188.70*** | ***21.05*** | ***22.96*** | ***1.91*** | ***209.75*** | ***211.66*** |
| ***22H*** | ***198.20*** | ***21.93*** | ***23.51*** | ***1.58*** | ***220.13*** | ***221.71*** |
| ***23H*** | ***207.70*** | ***23.20*** | ***24.84*** | ***1.64*** | ***230.90*** | ***232.54*** |
| ***24H*** | ***217.20*** | ***24.19*** | ***25.24*** | ***1.05*** | ***241.39*** | ***242.44*** |
| ***25H*** | ***226.70*** | ***25.38*** | ***27.93*** | ***2.55*** | ***252.08*** | ***254.63*** |
| ***26H*** | ***236.20*** | ***25.38*** | ***29.66*** | ***4.28*** | ***261.58*** | ***265.86*** |
| *27H* | *245.70* | *25.38* | *34.46* | *9.08* | *271.08* | *280.16* |
| **162-982B-** | | | | | | |
| 1H | 0.00 | 0.00 | 0.00 | 0.00 | 0.00 | 0.00 |
| 2H | 5.50 | 1.32 | 1.32 | 0.00 | 6.82 | 6.82 |
| 3H | 15.00 | 2.16 | 2.16 | 0.00 | 17.16 | 17.16 |
| 4H | 24.50 | 3.08 | 3.08 | 0.00 | 27.58 | 27.58 |
| 5H | 34.00 | 4.08 | 4.08 | 0.00 | 38.08 | 38.08 |
| 6H | 43.50 | 5.51 | 5.51 | 0.00 | 49.01 | 49.01 |
| 7H | 53.00 | 5.77 | 5.77 | 0.00 | 58.77 | 58.77 |
| 8H | 62.50 | 6.83 | 6.83 | 0.00 | 69.33 | 69.33 |
| 9H | 72.00 | 7.94 | 7.94 | 0.00 | 79.94 | 79.94 |
| 10H | 81.50 | 9.21 | 9.21 | 0.00 | 90.71 | 90.71 |
| 11H | 91.00 | 10.07 | 10.07 | 0.00 | 101.07 | 101.07 |
| 12H | 100.50 | 10.49 | 10.49 | 0.00 | 110.99 | 110.99 |
| 13H | 110.00 | 11.54 | 11.54 | 0.00 | 121.54 | 121.54 |
| *14H* | *119.50* | *12.60* | *12.50* | *-0.10* | *132.10* | *132.00* |
| ***15H*** | ***129.00*** | ***14.06*** | ***13.70*** | ***-0.36*** | ***143.06*** | ***142.70*** |
| *16H* | *138.50* | *14.74* | *14.45* | *-0.29* | *153.24* | *152.95* |
| *17H* | *148.00* | *15.59* | *15.44* | *-0.15* | *163.59* | *163.44* |
| *18H* | *157.50* | *16.58* | *16.43* | *-0.15* | *174.08* | *173.93* |
| ***19H*** | ***167.00*** | ***17.79*** | ***17.52*** | ***-0.27*** | ***184.79*** | ***184.52*** |
| ***20H*** | ***176.50*** | ***18.66*** | ***18.37*** | ***-0.29*** | ***195.16*** | ***194.87*** |
| ***21H*** | ***186.00*** | ***19.60*** | ***19.46*** | ***-0.14*** | ***205.60*** | ***205.46*** |
| ***22H*** | ***195.50*** | ***19.01*** | ***20.62*** | ***1.61*** | ***214.51*** | ***216.12*** |
| ***23H*** | ***205.00*** | ***20.42*** | ***21.96*** | ***1.54*** | ***225.42*** | ***226.96*** |
| ***24H*** | ***214.50*** | ***19.45*** | ***23.39*** | ***3.94*** | ***233.95*** | ***237.89*** |
| ***25H*** | ***224.00*** | ***20.59*** | ***25.09*** | ***4.50*** | ***244.59*** | ***249.09*** |
| ***26H*** | ***233.50*** | ***20.59*** | ***26.89*** | ***6.30*** | ***254.09*** | ***260.39*** |
| ***27X*** | ***243.00*** | ***20.59*** | ***28.42*** | ***7.83*** | ***263.59*** | ***271.42*** |
| ***28X*** | ***249.30*** | ***20.59*** | ***29.60*** | ***9.01*** | ***269.89*** | ***278.90*** |
| *29X* | *258.90* | *20.59* | *29.67* | *9.08* | *279.49* | *288.57* |
| *30X* | | *20.59* | *29.67* | *9.08* | | |
| *31X* | *278.20* | *20.59* | *29.67* | *9.08* | *298.79* | *307.87* |

| | | | | | | |
|---|---|---|---|---|---|---|
| 32X | 287.80 | 20.59 | 29.67 | 9.08 | 308.39 | 317.47 |
| 33X | 297.40 | 20.59 | 29.67 | 9.08 | 317.99 | 327.07 |
| 34X | 307.10 | 20.59 | 29.67 | 9.08 | 327.69 | 336.77 |
| 35X | 316.70 | 20.59 | 29.67 | 9.08 | 337.29 | 346.37 |
| 36X | 326.30 | 20.59 | 29.67 | 9.08 | 346.89 | 355.97 |
| 37X | 336.00 | 20.59 | 29.67 | 9.08 | 356.59 | 365.67 |
| 38X | 345.60 | 20.59 | 29.67 | 9.08 | 366.19 | 375.27 |
| 39X | 355.20 | 20.59 | 29.67 | 9.08 | 375.79 | 384.87 |
| 40X | 364.90 | 20.59 | 29.67 | 9.08 | 385.49 | 394.57 |
| 41X | 374.50 | 20.59 | 29.67 | 9.08 | 395.09 | 404.17 |
| 42X | 384.10 | 20.59 | 29.67 | 9.08 | 404.69 | 413.77 |
| 43X | 393.80 | 20.59 | 29.67 | 9.08 | 414.39 | 423.47 |
| 44X | 403.40 | 20.59 | 29.67 | 9.08 | 423.99 | 433.07 |
| 45X | 413.00 | 20.59 | 29.67 | 9.08 | 433.59 | 442.67 |
| 46X | 422.60 | 20.59 | 29.67 | 9.08 | 443.19 | 452.27 |
| 47X | 432.20 | 20.59 | 29.67 | 9.08 | 452.79 | 461.87 |
| 48X | 441.80 | 20.59 | 29.67 | 9.08 | 462.39 | 471.47 |
| 49X | 451.40 | 20.59 | 29.67 | 9.08 | 471.99 | 481.07 |
| 50X | 461.00 | 20.59 | 29.67 | 9.08 | 481.59 | 490.67 |
| 51X | 470.60 | 20.59 | 29.67 | 9.08 | 491.19 | 500.27 |
| 52X | 480.20 | 20.59 | 29.67 | 9.08 | 500.79 | 509.87 |
| 53X | 489.70 | 20.59 | 29.67 | 9.08 | 510.29 | 519.37 |
| 54X | 499.30 | 20.59 | 29.67 | 9.08 | 519.89 | 528.97 |
| 55X | 508.90 | 20.59 | 29.67 | 9.08 | 529.49 | 538.57 |
| 56X | 518.50 | 20.59 | 29.67 | 9.08 | 539.09 | 548.17 |
| 57X | 528.10 | 20.59 | 29.67 | 9.08 | 548.69 | 557.77 |
| 58X | 537.70 | 20.59 | 29.67 | 9.08 | 558.29 | 567.37 |
| 59X | | 20.59 | 29.67 | 9.08 | | |
| 60X | 557.00 | 20.59 | 29.67 | 9.08 | 577.59 | 586.67 |
| 61X | 566.70 | 20.59 | 29.67 | 9.08 | 587.29 | 596.37 |
| 62X | 576.40 | 20.59 | 29.67 | 9.08 | 596.99 | 606.07 |
| 63X | | 20.59 | 29.67 | 9.08 | | |
| 64X | | 20.59 | 29.67 | 9.08 | | |
| 65X | | 20.59 | 29.67 | 9.08 | | |
| **162-982C-** | | | | | | |
| 1H | 0.00 | 0.15 | 0.15 | 0.00 | 0.15 | 0.15 |
| 2H | 3.80 | -0.80 | 0.80 | 1.60 | 3.00 | 4.60 |
| 3H | 13.30 | -0.02 | 0.02 | 0.04 | 13.28 | 13.32 |
| 4H | 22.80 | 3.52 | 3.52 | 0.00 | 26.32 | 26.32 |
| 5H | 32.30 | 3.26 | 3.26 | 0.00 | 35.56 | 35.56 |
| 6H | 41.80 | 4.58 | 4.58 | 0.00 | 46.38 | 46.38 |
| 7H | 51.30 | 5.65 | 5.65 | 0.00 | 56.95 | 56.95 |
| 8H | 60.80 | 5.79 | 5.79 | 0.00 | 66.59 | 66.59 |
| 9H | 70.30 | 5.82 | 5.82 | 0.00 | 76.12 | 76.12 |
| 10H | 79.80 | 7.19 | 7.19 | 0.00 | 86.99 | 86.99 |
| 11H | 89.30 | 8.17 | 8.17 | 0.00 | 97.47 | 97.47 |
| 12H | 98.80 | 9.05 | 9.05 | 0.00 | 107.85 | 107.85 |
| 13H | 108.30 | 10.45 | 10.45 | 0.00 | 118.75 | 118.75 |
| 14H | 117.80 | 11.13 | 11.13 | 0.00 | 128.93 | 128.93 |
| **15H** | **127.30** | **13.08** | **12.92** | **-0.16** | **140.38** | **140.22** |
| **16H** | **136.80** | **13.88** | **13.52** | **-0.36** | **150.68** | **150.32** |
| **17H** | **146.30** | **14.45** | **14.30** | **-0.15** | **160.75** | **160.60** |
| **18H** | **155.80** | **15.89** | **15.74** | **-0.15** | **171.69** | **171.54** |
| **19H** | **165.30** | **16.43** | **16.16** | **-0.27** | **181.73** | **181.46** |
| **20H** | **174.80** | **18.41** | **17.97** | **-0.44** | **193.21** | **192.77** |
| **21H** | **184.30** | **19.90** | **18.95** | **-0.95** | **204.20** | **203.25** |
| **22H** | **193.80** | **18.88** | **20.39** | **1.51** | **212.68** | **214.19** |
| 23H | 203.30 | 19.79 | 21.33 | 1.54 | 223.09 | 224.63 |
| 24H | 212.80 | 19.60 | 23.54 | 3.94 | 232.40 | 236.34 |
| **25H** | **222.30** | **20.66** | **25.23** | **4.57** | **242.96** | **247.53** |
| 26H | 231.80 | 20.66 | 26.96 | 6.30 | 252.46 | 258.76 |
| **27H** | **241.30** | **20.66** | **29.74** | **9.08** | **261.96** | **271.04** |
| **162-982D-** | | | | | | |
| 1H | 20 | 2.18 | 2.18 | 0.00 | 22.18 | 22.18 |

| Table 2: Revised Splice for ODP Site 982 | | | | | | |
|---|---|---|---|---|---|---|
| *Key* | **BOLD = new splice tie point** | | | *Italics = old splice tie point with revised composite depth* | | |
| **Upper core tie point** | | | | **Lower core tie point ID** | | |
| **ID** | **Depth** | | | **ID** | **Depth** | |
| *Hole, core, section, interval (cm)* | *(mbsf [m CSF-A])* | *(rmcd [m revised CCSF-A])* | | *Hole, core, section, interval (cm)* | *(mbsf [m CSF-A])* | *(rmcd [m revised CCSF-A])* |
| **162-** | | | | **162-** | | |
| U982A-1H-6, 18 | 7.68 | 7.68 | Tie to | U982B-2H-1, 86 | 6.36 | 7.68 |
| U982B-2H-7, 2 | 14.52 | 15.84 | Tie to | U982C-3H-2, 106 | 15.86 | 15.84 |
| U982C-3H-4, 90 | 18.7 | 18.68 | Tie to | U982B-3H-2, 2 | 16.52 | 18.68 |
| U982B-3H-5, 140 | 22.4 | 24.56 | Tie to | U982D-1H-2, 120 | 22.63 | 24.56 |
| U982D-1H-5, 29 | 26.22 | 28.15 | Tie to | U982B-4H-1, 57 | 25.07 | 28.15 |
| U982B-4H-6, 74 | 32.74 | 35.82 | Tie to | U982C-5H-1, 26 | 32.56 | 35.82 |
| U982C-5H-3, 58 | 35.88 | 39.14 | Tie to | U982B-5H-1, 106 | 35.06 | 39.14 |
| U982B-5H-6, 90 | 42.4 | 46.48 | Tie to | U982C-6H-1, 10 | 41.9 | 46.48 |
| U982C-6H-3, 66 | 45.46 | 50.04 | Tie to | U982B-6H-1, 103 | 44.53 | 50.04 |
| U982B-6H-6, 98 | 51.98 | 57.49 | Tie to | U982A-6H-4, 26 | 50.96 | 57.49 |
| U982A-6H-5, 98 | 53.18 | 59.71 | Tie to | U982B-7H-1, 94 | 53.94 | 59.71 |
| U982B-7H-5, 146 | 60.46 | 66.23 | Tie to | U982A-7H-2, 106 | 58.26 | 66.23 |
| U982A-7H-4, 138 | 61.58 | 69.55 | Tie to | U982B-8H-1, 22 | 62.72 | 69.55 |
| U982B-8H-5, 3 | 68.53 | 75.36 | Tie to | U982A-8H-1, 138 | 66.58 | 75.36 |
| U982A-8H-5, 98 | 72.18 | 80.96 | Tie to | U982B-9H-1, 102 | 73.02 | 80.96 |
| U982B-9H-6, 131 | 80.81 | 88.75 | Tie to | U982C-10H-2, 26 | 81.56 | 88.75 |
| U982C-10H-7, 4 | 88.84 | 96.03 | Tie to | U982B-10H-4, 82 | 86.82 | 96.03 |
| U982B-10H-6, 26 | 89.26 | 98.47 | Tie to | U982A-10H-2, 134 | 87.04 | 98.47 |
| U982A-10H-6, 82 | 92.52 | 103.95 | Tie to | U982B-11H-2, 138 | 93.88 | 103.95 |
| U982B-11H-6, 74 | 99.24 | 109.31 | Tie to | U982A-11H-3, 26 | 96.96 | 109.31 |
| U982A-11H-5, 24 | 99.94 | 112.29 | Tie to | U982B-12H-1, 130 | 101.8 | 112.29 |
| U982B-12H-7, 12 | 109.62 | 120.11 | Tie to | U982A-12H-3, 42 | 106.62 | 120.11 |
| U982A-12H-5, 125 | 110.45 | 123.94 | Tie to | U982B-13H-2, 90 | 112.4 | 123.94 |
| *U982B-13H-6, 74* | *118.24* | *129.78* | *Tie to* | *U982A-13H-2, 76* | *114.96* | *129.78* |
| *U982A-13H-4, 122* | *118.42* | *133.24* | *Tie to* | *U982B-14H-1, 124* | *120.74* | *133.24* |
| *U982B-14H-7, 34* | *128.44* | *140.94* | *Tie to* | *U982C-15H-1, 72* | *128.02* | *140.94* |
| *U982C-15H-3, 26* | *130.56* | *143.48* | *Tie to* | *U982B-15H-1, 78* | *129.78* | *143.48* |
| *U982B-15H-6, 50* | *137* | *150.7* | *Tie to* | *U982A-15H-2, 51* | *133.71* | *150.7* |
| *U982A-15H-5, 76* | *138.44* | *155.43* | *Tie to* | *U982B-16H-2, 98* | *140.98* | *155.43* |
| *U982B-16H-6, 82* | *146.82* | *161.27* | *Tie to* | *U982A-16H-2, 60* | *143.3* | *161.27* |
| *U982A-16H-6, 84* | *149.54* | *167.51* | *Tie to* | *U982B-17H-3, 107* | *152.07* | *167.51* |
| *U982B-17H-6, 18* | *155.68* | *171.12* | *Tie to* | *U982A-17H-2, 7* | *152.27* | *171.12* |
| *U982A-17H-6, 51* | *158.71* | *177.56* | *Tie to* | *U982B-18H-3, 63* | *161.13* | *177.56* |
| **U982B-18H-6, 66** | **165.66** | **182.09** | **Tie to** | **U982A-18H-2, 104** | **162.74** | **182.09** |
| **U982A-18H-6, 52** | **168.22** | **187.57** | **Tie to** | **U982B-19H-3, 5** | **170.05** | **187.57** |
| **U982B-19H-7, 8** | **176.08** | **193.6** | **Tie to** | **U982C-20H-1, 83** | **175.63** | **193.6** |
| *U982C-20H-4, 7* | *179.37* | *197.34* | *Tie to* | *U982B-20H-2, 97* | *178.97* | *197.34* |
| **U982B-20H-6, 2** | **184.02** | **202.39** | **Tie to** | **U982A-20H-1, 149** | **180.69** | **202.39** |
| **U982A-20H-6, 53** | **187.23** | **208.93** | **Tie to** | **U982B-21H-3, 47** | **189.47** | **208.93** |
| **U982B-21H-6, 42** | **193.9** | **213.36** | **Tie to** | **U982A-21H-2, 20** | **190.4** | **213.36** |
| **U982A-21H-6, 81** | **197.01** | **219.97** | **Tie to** | **U982C-22H-4, 128** | **199.58** | **219.97** |
| **U982C-22H-6, 91** | **202.21** | **222.6** | **Tie to** | **U982A-22H-1, 89** | **199.09** | **222.6** |
| **U982A-22H-5, 111** | **205.31** | **228.82** | **Tie to** | **U982B-23H-2, 36** | **206.86** | **228.82** |
| **U982B-23H-5, 112** | **212.12** | **234.08** | **Tie to** | **U982A-23H-2, 4** | **209.24** | **234.08** |
| **U982A-23H-6, 12** | **215.32** | **240.16** | **Tie to** | **U982B-24H-2, 77** | **216.77** | **240.16** |
| **U982B-24H-5, 39** | **220.89** | **244.28** | **Tie to** | **U982A-24H-2, 34** | **219.04** | **244.28** |
| **U982A-24H-6, 33** | **225.03** | **250.27** | **Tie to** | **U982B-25H-1, 118** | **225.18** | **250.27** |
| **U982B-25H-5, 136** | **231.36** | **256.45** | **Tie to** | **U982A-25H-2, 32** | **228.52** | **256.45** |
| **U982A-25H-6, 30** | **234.5** | **262.43** | **Tie to** | **U982B-26H-2, 54** | **235.54** | **262.43** |
| **U982B-26H-5, 101** | **240.51** | **267.4** | **Tie to** | **U982A-26H-2, 4** | **237.74** | **267.4** |
| **U982A-26H-6, 33** | **244.03** | **273.69** | **Tie to** | **U982C-27H-2, 115** | **243.95** | **273.69** |
| **U982C-27H-7, 4** | **250.34** | **280.08** | **Tie to** | **U982B-28X-1, 118** | **250.48** | **280.08** |
| *U982B-62X-1, 39* | *576.79* | *597.38* | | | | |

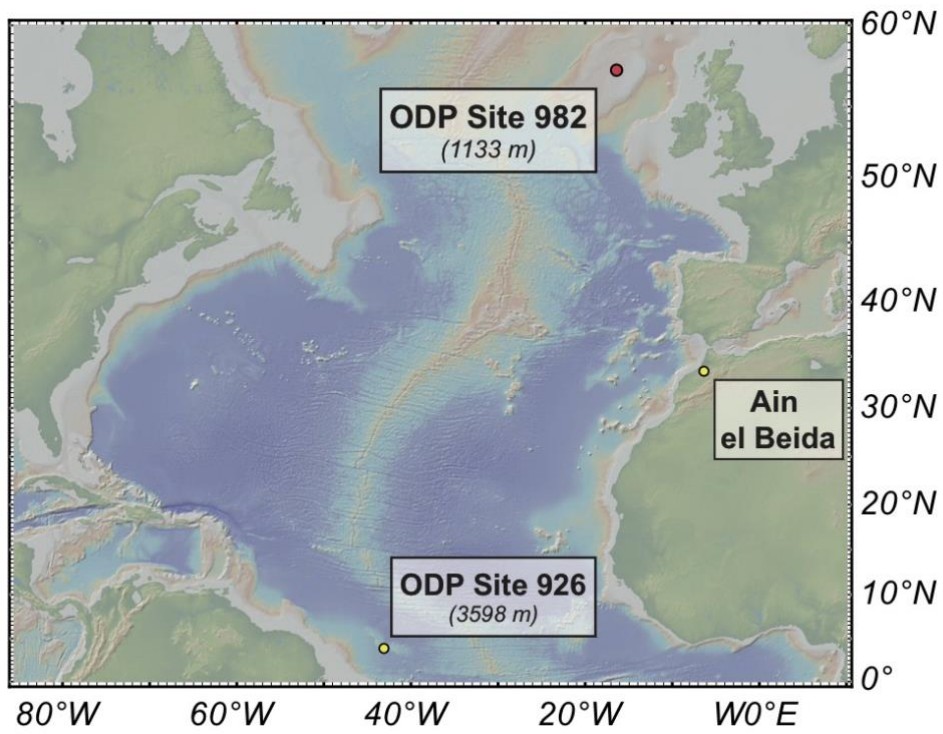

**Figure 1: Map showing the location of ODP Sites 982 and 926, as well as the Ain el Beida quarry section. Red site locations indicate new data was produced, yellow site locations indicate the use of published data.**

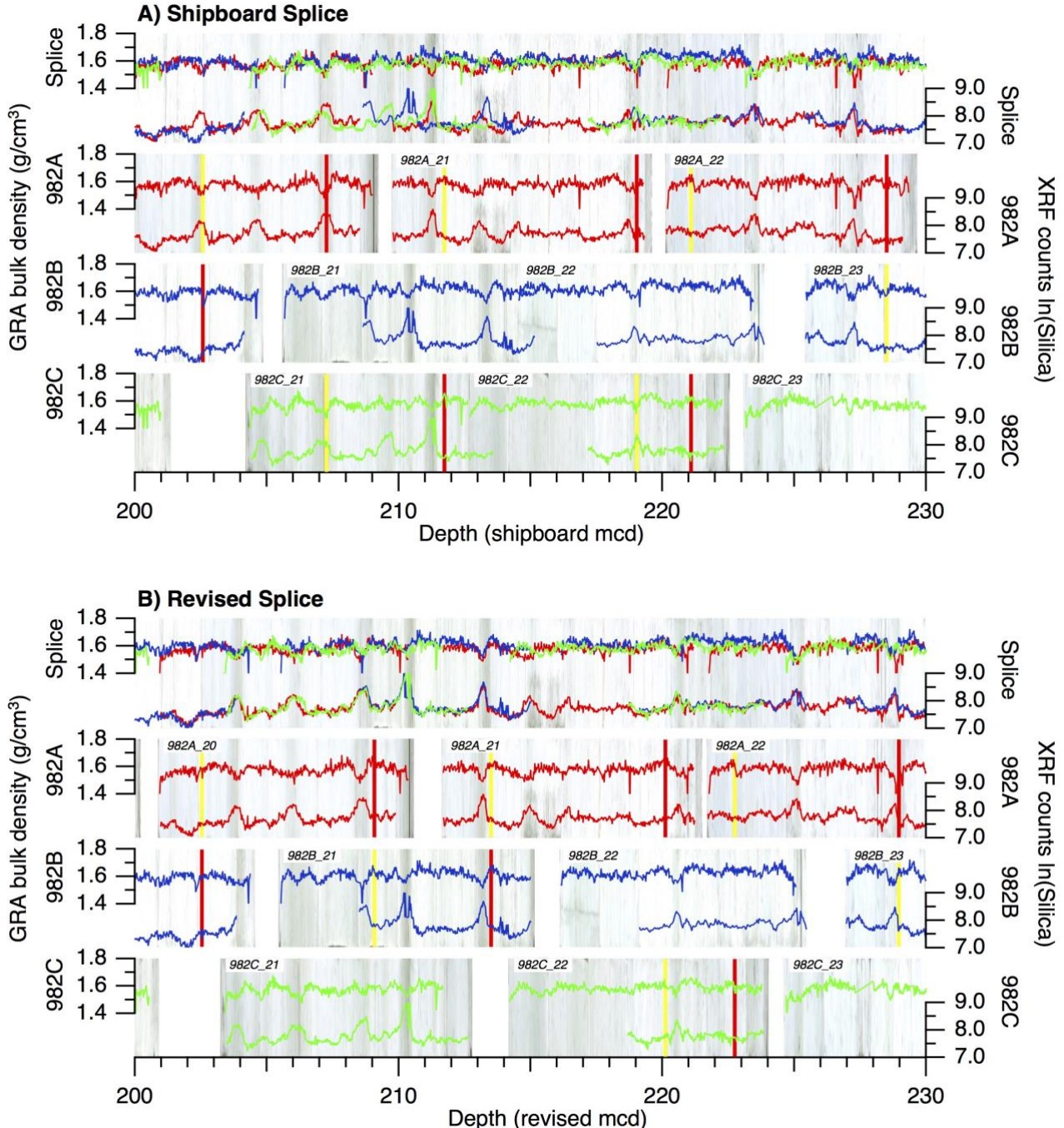

**Figure 2: Panels showing A) the shipboard composite splice and B) the revised splice between 200 and 230 m composite depth. Each panel consists of the composite core images of the Site 982 splice and Holes 982A, 982B and 982C, with the shipboard GRA bulk density data and the new XRF ln(Silica counts) record. Tie point locations are shown by yellow (upper tie point within a core) and red (lower tie point within a core) lines. See Supplementary Figure 2 for images of the entire revised splice interval. We could not include the reflectance data, since it was not present in the ODP database, but instead was archived as damaged data.**

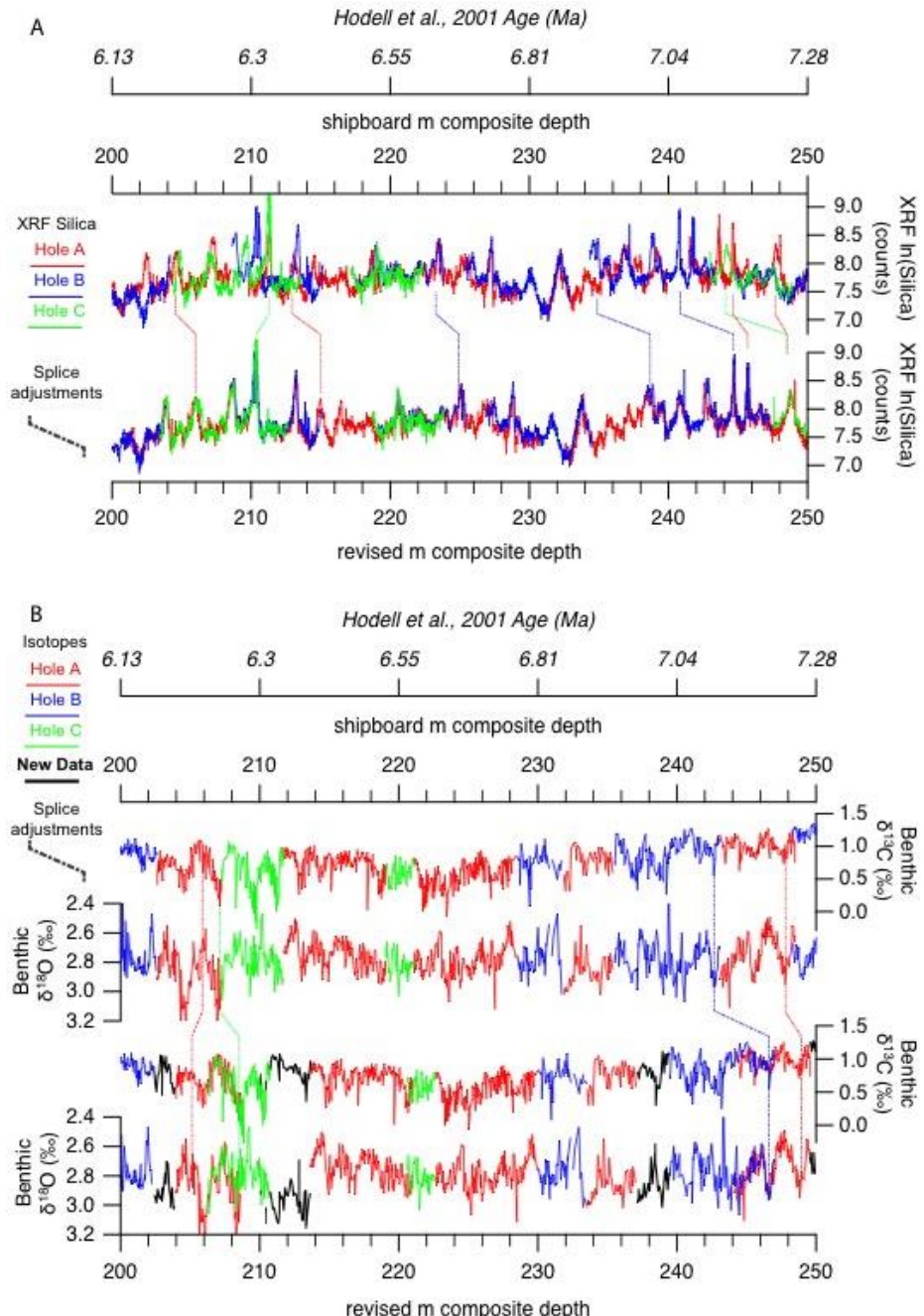

**Figure 3: A) XRF ln(Silica counts) for the individual holes at Site 982 on the shipboard composite depth with the corresponding Hodell et al. (2001) age on the top axis, and on the revised composite depth on the bottom axis. B) Stable isotope $\delta^{13}C$ and $\delta^{18}O$ records for the individual holes at Site 982 published in Hodell et al. (2001) on the shipboard composite depth with the corresponding age on the top axis. On the bottom axis, the published stable isotope data is plotted on the revised composite depth, together with the new stable isotope data (black lines) produced in this study. Vertical lines show how the splice was adjusted and highlight the effect of these adjustments on the agreement between the XRF data from the three holes. See Supplementary Figure 3 for plots of the entire revised splice interval.**

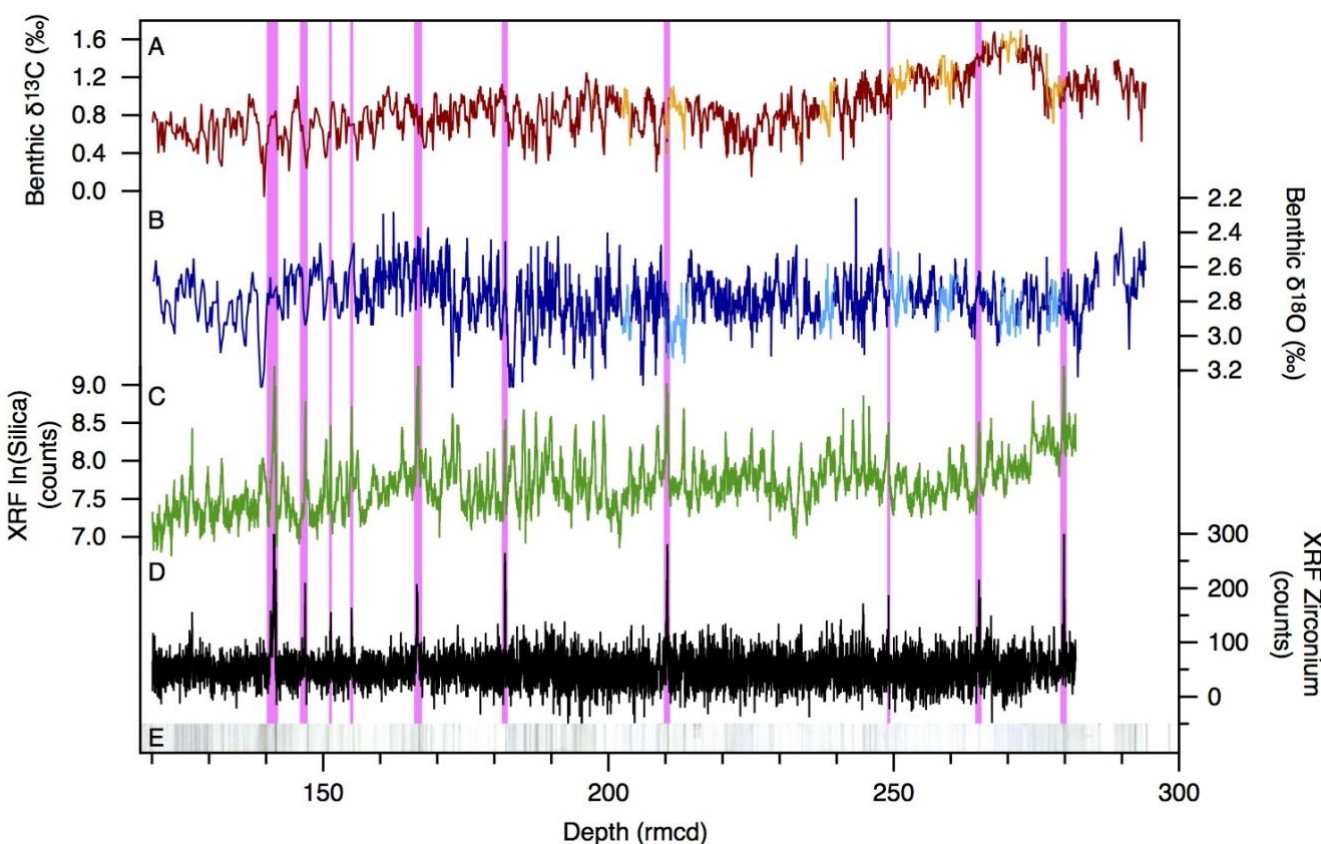

**Figure 4: Overview of the new XRF and updated stable isotope datasets on the revised composite depth at Site 982. The complete benthic A) δ¹³C records from this study (orange) and Hodell et al. (2001) (red) and B) δ¹⁸O records from this study (light blue) and Hodell et al. (2001) (dark blue), the XRF counts for C) ln(Silica) (green) and D) Zirconium (Zr) (black), and E) the composite splice image for Site 982. Pink intervals highlight the location of ash layers, based on the presence of Zr peaks, which affect the XRF data, but show no influence on the stable isotope records.**

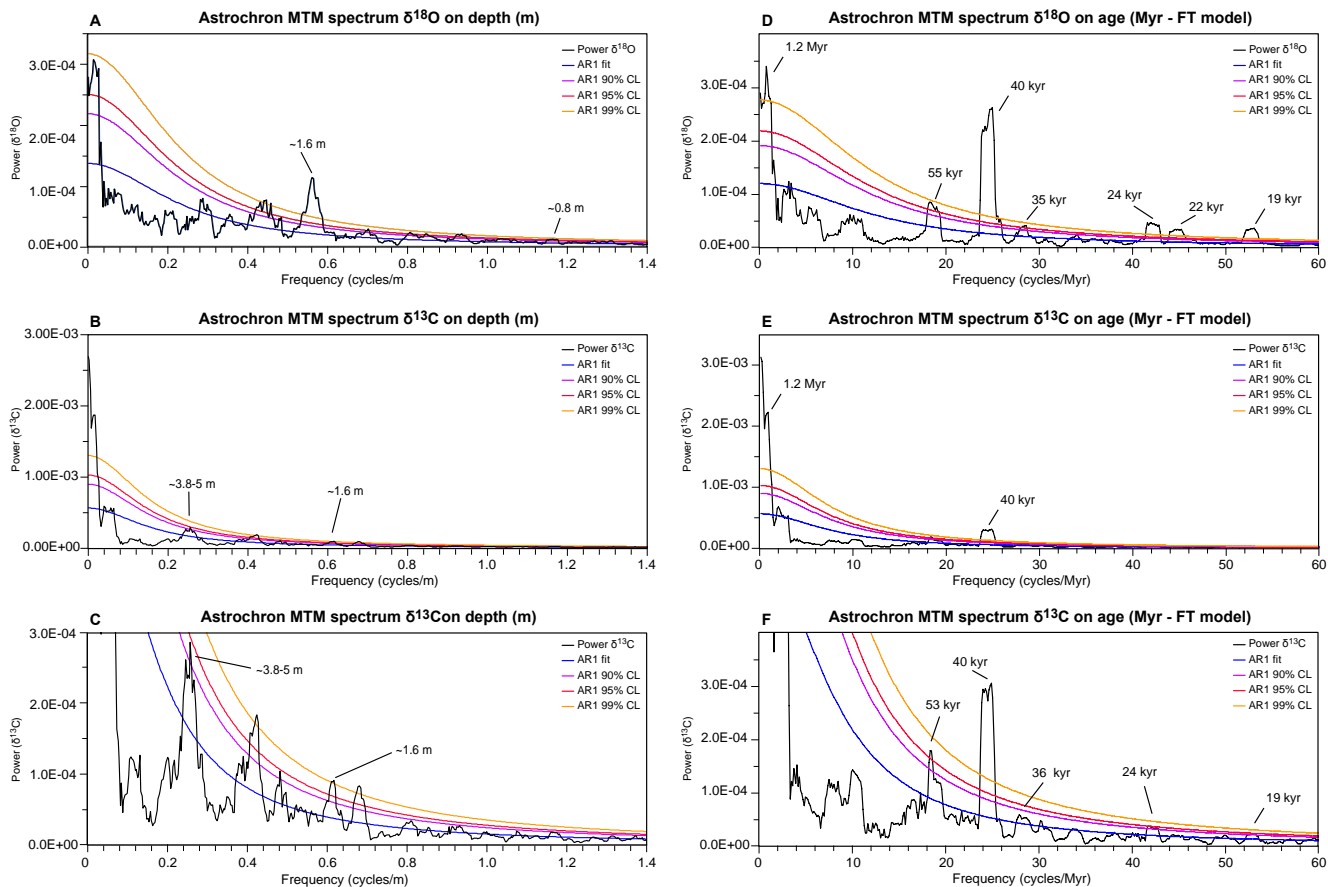

**Figure 5: Multitaper method (MTM) power spectra of the entire stable isotope stratigraphies (8.0-4.5 Ma) generated using Astrochron (Meyers, 2014; for the specific code, please see the Supplementary Information) for δ¹⁸O in A)/D) and δ¹³C with a full vertical axis shown in B)/E) and vertically expanded in C)/F) to show the same vertical axis extent as in A)/D. The red noise fit (AR1) and 90, 95 and 99% confidence limits (CL) relative to red noise are also indicated. The periods of different peaks are annotated. The MTM spectra are shown in the depth domain in A), B) and C) and in the time domain in D) E) and F).**

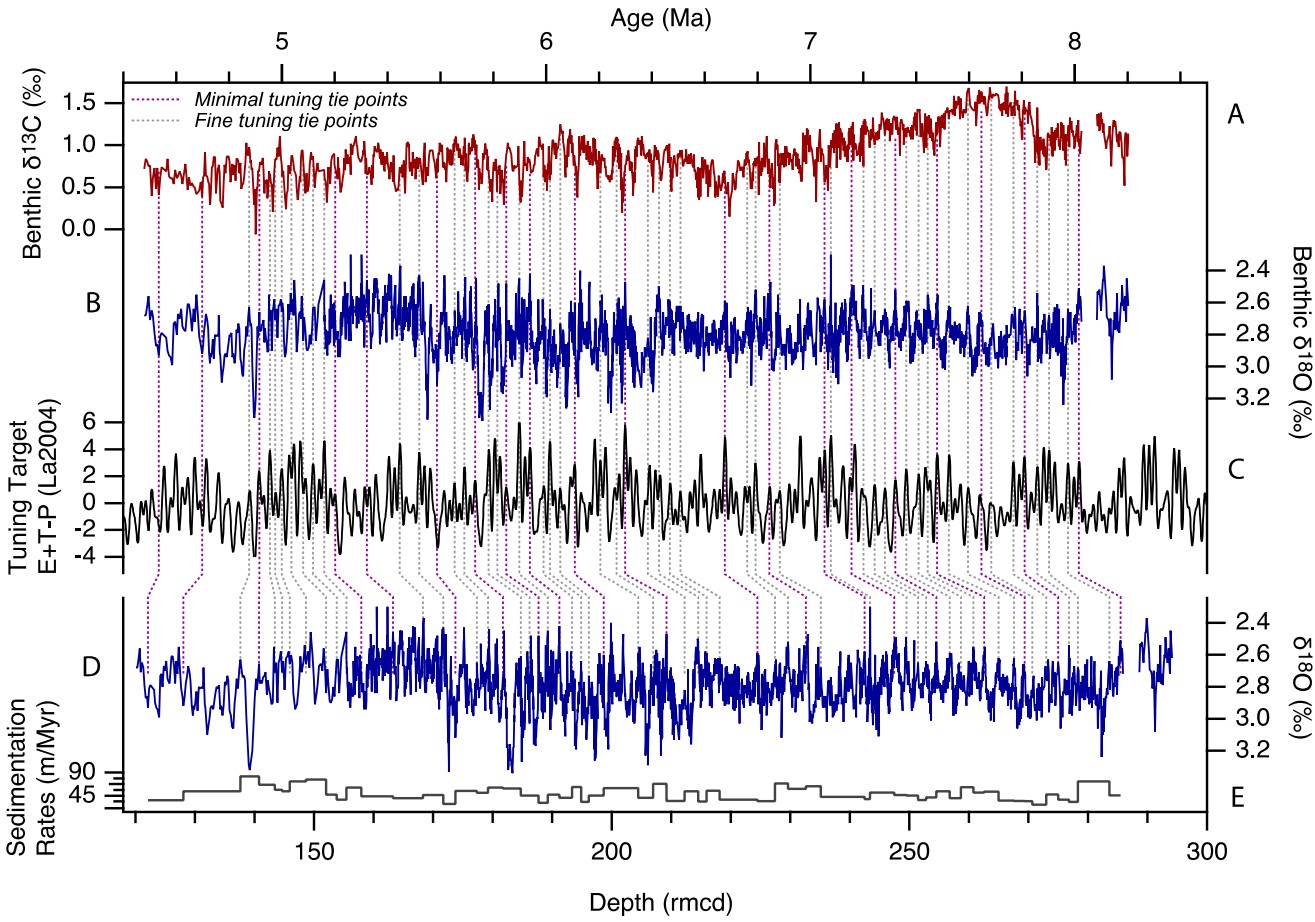

**Figure 6: Fine-tuned astrochronology for Site 982, with minimal tuning tie points indicated in purple and fine-tuning tie points indicated in grey. A) Fine-tuned benthic foraminiferal δ¹³C (in ‰ versus VPDB). B) Fine-tuned benthic foraminiferal δ¹⁸O (in ‰ versus VPDB). C) Eccentricity+Tilt-Precession tuning target (E+T-P) from Laskar et al. (2004) D) Benthic foraminiferal δ¹⁸O (in ‰ versus VPDB) on depth rmcd. E) Fine-tuned sedimentation rates (in m/Myr) on depth rmcd.**

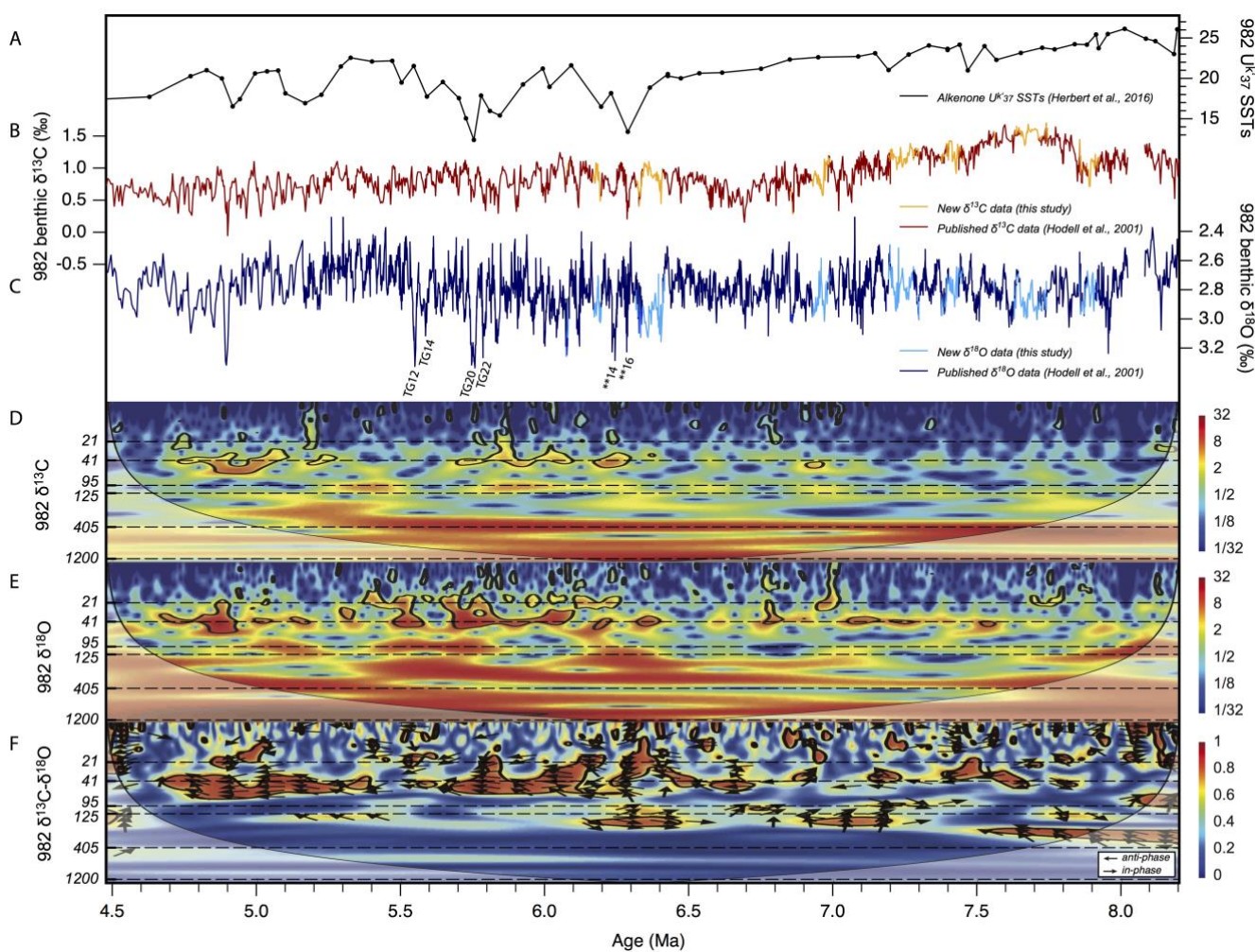

**Figure 7: A) Alkenone U$^{k'}_{37}$-derived SSTs from Site 982 (Herbert et al., 2016). B) Fine-tuned Site 982 benthic foraminiferal δ$^{13}$C (in ‰ versus VPDB). C) Fine-tuned Site 982 benthic foraminiferal δ$^{18}$O (in ‰ versus VPDB). D) Wavelet analysis of the Site 982 benthic δ$^{13}$C data (in ‰ versus VPDB) on the fine tuned astrochronology. E) Wavelet analysis of the Site 982 benthic δ$^{18}$O data (in ‰ versus VPDB) on the fine tuned astrochronology. F) Coherence wavelet analysis of the Site 982 benthic δ$^{13}$C and δ$^{18}$O, with the black arrows to the left indicating anti-phase and the black arrows to the right indicating in-phase. Prominent MIS are annotated, with ** standing for C3An.δ$^{18}$O. All wavelets were generated using Grinsted et al. (2004).**

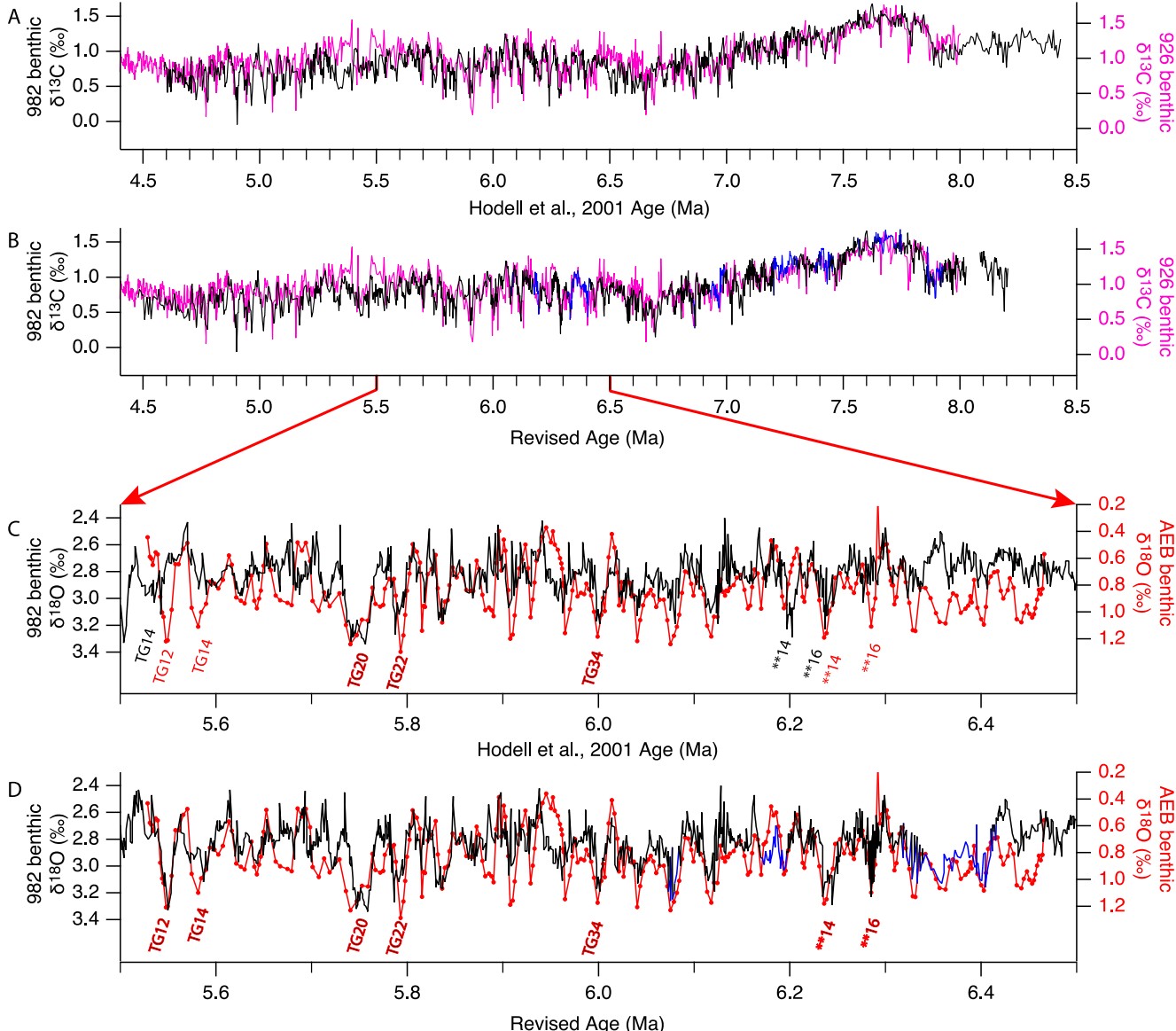

**Figure 8: A)** The benthic δ¹³C record from ODP Site 982 published in Hodell et al. (2001) on its original age model, compared to the benthic δ¹³C record from ODP Site 926 on its astronomical age model (Drury et al., 2017; Wilkens et al., 2017; Zeeden et al., 2013). **B)** The revised benthic δ¹³C record, with new data indicated in blue, on the revised age model, compared to the benthic δ¹³C record from ODP 926 on its astronomical age model (Drury et al., 2017; Wilkens et al., 2017; Zeeden et al., 2013). **C)** The benthic δ¹⁸O record from ODP Site 982 published in Hodell et al. (2001) on its original age model, compared to the benthic δ¹⁸O record from Ain El Beida on its astronomical age model (van der Laan et al., 2005, 2012). **D)** The revised benthic δ¹⁸O record, with new data indicated in blue, on the revised age model, compared to the benthic δ¹⁸O record from Ain El Beida on its astronomical age model (van der Laan et al., 2005, 2012). In panels C) and D), MIS are indicated as they were originally identified in by Hodell et al. (2001) in black and by van der Laan et al. (2005) in red. Where the identification of an MIS agrees, either in the original or revised chronology, the stage is indicated in two-tone red and black. The ** is short for C3An.δ¹⁸O (see van der Laan et al. (2005) for further details).