# Peer review of "Reinforcing the North Atlantic backbone: revision and extension of the composite splice at ODP Site 982"

_Climate of the Past, 2017_

## Referee Comment (RC1) · Anonymous Referee #1 · 7 Oct 2017

A very well written and easy to follow Ms. I have no issues with your conclusions, which seem robust and exciting to me. In addition to the minor issues I raise below, I think the impact of your Ms would be greater if you were to incorporate into your figures the original GRAPE and spectral reflectance data used to define the shipboard splice. This way you could illustrate more clearly how what seemed like decent splicing at the time has been shown to require revision with XRF-core scanning. I don't think this finding is surprising to a stratigrapher (it's just part of how we can build on and improve high-quality shipboard work, post-cruise), but it would represent a more impactful lesson for. e.g., proxy users that do not necessarily consider splice/age-model robustness when interpreting their data.

[Figure]

Minor comments: Page 3, line 10: its not the site's proximity to the Med Sea that makes it important for correlation to the Med, it's the site's location in the North Atlantic that makes it important for correlation to Med records.

Page 4, lines 4-14: presumably you generated benthic d18O data from Holes A, B and C from stratigraphic gaps identified in your new splice and from portions in the existing shipboard splice (to verify no offsets in values). I think it is necessary to make that clearer in your methods text here. I know I can look at Table 3, but make it easier for the reader to understand. Saying generically that you measured 263 new data between 200 and 280 mcd is not informative enough. Having now read on I can see that the statement starting on page 5, line 4 needs folding into your methods here.

Page 4, line 28: I would like to see the rmcd splice Tables 4 and 5 in the main text. It's a hard enough job convincing people that stratigraphy is vitally important and that all subsequent results are potentially erroneous if your splice and age model is wrong. This sort of study underlies why proxy users should not blindly use published stratigraphies and having the new splice tables as supplementary tables makes stratigraphy feel like just that, supplementary.

Page 5, line 9: it would be useful for you to show evidence for the existence of these strong cycles in benthic d18O and d13C and their anti-phase relationship, beyond visual inspection (show results of TSA in the depth-domain for starters?).

Page 5, line 31: replace 'as' with 'since'.

Page 6, line 18: please include the Drury et al. Site U1337 record in Figure 6. I'm pleased that you comment on the timing of your obliquity-dominated intervals and amplitude changes in benthic d18O with the identification SST cooling to near-modern values between about 7 and 5.4 Mya by Herbert et al. (2016). Really nice finding!

Page 8, Line 31-32: regardless of the completeness of the AEB record, as you say, the absence of cycles in this record - otherwise present in your revised 982 splice - seems

most likely be due to lack of data resolution at the former.

Page 9, lines 6-14: I would fold this important statement about MIS revision into your conclusions (and abstract) and also condense your conclusions section, which could be shorter.

Figure 4: caption needs an '(e)' inserted after 'the composite splice image for Site 982'. Also, A-E labels in the figure care capitalised, but they are lowercase in the caption. This is an issue for all figures that you may wish to rectify.

Figure 5: would be nice to have labels directly on the figure to aid the reader in quicker identification of minimal- and fine-tuning tie points without having to read the caption. Age scale on top x-axis is missing a label. I would label stable isotope data as 'benthic' on each y-axis label. Something to correct for all figures? I know its stated in the caption, but the first thing one looks at are the data in the figure!

Figure 8: red arrows showing link between 982/926 records and zoomed in interval for the AEB record are messy since they cover the numbers on the x-axis of panel B. Please redesign. I think Figure 8 could be redesigned as follows: panel A = revised 982 with 926, panel B = original 982 stratigraphy/record, panel C = original 982 with AEB, panel D = revised 982 record with AEB. Just a suggestion.

---

## Author Comment (AC1) · 15 Nov 2017

Dear reviewer,

Thank you very much for your constructive review. Your feedback was very helpful and will improve our manuscript. Our replies to your comments (*in italics*) are shown below in red.

*A very well written and easy to follow Ms. I have no issues with your conclusions, which seem robust and exciting to me. In addition to the minor issues I raise below, I think the impact of your Ms would be greater if you were to incorporate into your figures the original GRAPE and spectral reflectance data used to define the shipboard splice. This way you could illustrate more clearly how what seemed like decent splicing at the time has been shown to require revision with XRF-core scanning. I don't think this finding is surprising to a stratigrapher (it's just part of how we can build on and improve high- quality shipboard work, post-cruise), but it would represent a more impactful lesson for. e.g., proxy users that do not necessarily consider splice/age-model robustness when interpreting their data.*
Thank you for this suggestion, we will incorporate the original GRAPE and spectral reflectance data into one of the figures, most likely into Figure 2 or 3. We will additionally add supplementary information figures showing these records for the entire studied interval of Site 982. As the reviewer points out, in addition to providing splice revisions for Site 982, it is also our intention to illustrate to all ODP/IODP users that they should consider how robust shipboard splices are and verifying them if they wish to generate high-resolution composite records.

*Minor comments:*
*Page 3, line 10: it's not the site's proximity to the Med Sea that makes it important for correlation to the Med, it's the site's location in the North Atlantic that makes it important for correlation to Med records.*
We agree and rephrased the sentence as follows: "Considering the site's location in the North Atlantic, this isotope record is important for comparing to isotopic records from the Mediterranean basin".

*Page 4, lines 4-14: presumably you generated benthic d18O data from Holes A, B and C from stratigraphic gaps identified in your new splice and from portions in the existing shipboard splice (to verify no offsets in values). I think it is necessary to make that clearer in your methods text here. I know I can look at Table 3, but make it easier for the reader to understand. Saying generically that you measured 263 new data between 200 and 280 mcd is not informative enough. Having now read on I can see that the statement starting on page 5, line 4 needs folding into your methods here.*
We indeed employed the strategy outlined here by the reviewer, measuring stable isotope data in the stratigraphic gaps, as well as overlapping with the original data to ensure the datasets could be integrated. We will clarify that in this section and move Line 4, Page 5 to Section 2.4.

*Page 4, line 28: I would like to see the rmcd splice Tables 4 and 5 in the main text. It's a hard enough job convincing people that stratigraphy is vitally important and that all subsequent results are potentially erroneous if your splice and age model is wrong. This sort of study underlies why proxy users should not blindly use published stratigraphies and having the new splice tables as supplementary tables makes stratigraphy feel like just that, supplementary.*
It was definitely not our intention to display the new offsets and splice as supplementary. We included them as supplementary excel tables as opposed to print tables to make it as easy as possible for users to extract the required information from them. However, we very much

appreciate the reviewer's comment and will include a print copy of the tables, as well as excel files in the supplement/PANGAEA to facilitate easy use.

*Page 5, line 9: it would be useful for you to show evidence for the existence of these strong cycles in benthic d18O and d13C and their anti-phase relationship, beyond visual inspection (show results of TSA in the depth-domain for starters?).*
We provide evidence of the strong cycles in the d18O and d13C records in the time domain in the form of wavelet and spectral analysis in both Figure 6 and Figure 7. However, we will also add spectral analysis for d18O and d13C in the depth domain to Figure 7 and add a coherency wavelet analysis in the time domain to Figure 6.

*Page 5, line 31: replace 'as' with 'since'.*
We will replace this.

*Page 6, line 18: please include the Drury et al. Site U1337 record in Figure 6. I'm pleased that you comment on the timing of your obliquity-dominated intervals and amplitude changes in benthic d18O with the identification SST cooling to near-modern values between about 7 and 5.4 Mya by Herbert et al. (2016). Really nice finding!*
We did not plot the Site U1337 benthic d18O data together with the 982 d18O data in this instance, as the compilation of the benthic d18O available for this interval is the focus of an in prep manuscript and we did not wish to detract from this ongoing work.

*Page 8, Line 31-32: regardless of the completeness of the AEB record, as you say, the absence of cycles in this record - otherwise present in your revised 982 splice – seems most likely be due to lack of data resolution at the former.*
We agree that the absence of cycles in the record is more likely due to the lower data resolution at AEB. We will clarify this further in the text by adding that low sampling resolution, in addition to low sedimentation rates, at AEB, are most likely the cause of the remaining disagreement between AEB and Site 982.

*Page 9, lines 6-14: I would fold this important statement about MIS revision into your conclusions (and abstract) and also condense your conclusions section, which could be shorter.*
We will incorporate this statement into both the abstract and conclusions and make the conclusions more succinct.

*Figure 4: caption needs an '(e)' inserted after 'the composite splice image for Site 982'. Also, A-E labels in the figure care capitalised, but they are lowercase in the caption. This is an issue for all figures that you may wish to rectify.*
We will rectify the omission of the (e) and change the labels to upper case letters in all captions.

*Figure 5: would be nice to have labels directly on the figure to aid the reader in quicker identification of minimal- and fine-tuning tie points without having to read the caption. Age scale on top x-axis is missing a label. I would label stable isotope data as 'benthic' on each y-axis label. Something to correct for all figures? I know its stated in the caption, but the first thing one looks at are the data in the figure!*
The labels for the minimal and fine tuning are on the figure in the bottom right, although we agree that they are not displayed in the most prominent position. We will include them in the top left of the figure above the benthic d13C data, within the box frame of the figure itself. We will also correct the missing age label and add "benthic" to the relevant y-axes.

*Figure 8: red arrows showing link between 982/926 records and zoomed in interval for the AEB record are messy since they cover the numbers on the x-axis of panel B. Please redesign. I think Figure 8 could be redesigned as follows: panel A = revised 982 with 926, panel B = original 982 stratigraphy/record, panel C = original 982 with AEB, panel D = revised 982 record with AEB. Just a suggestion.*

The red arrows unfortunately shifted at some stage during the figure making process and we only noticed this following the reviewer's comment. They should go from 5.5 and 6.5 and originally did not overlap with the numbers on the x-axis. We will rectify this in the revised version, either by shifting the arrows to the correct position or by removing them entirely. Additionally, the Hodell et al., 2001 Age (Ma) caption will be added to the A x-axis, as this was also accidentally deleted.

We appreciate the reviewer's suggestion for redesigning the figure, but we prefer keeping a consistent order for "original age" on the upper panel and "revised age" on the lower panel of the A/B and C/D sets.

---

## Referee Comment (RC2) · Anonymous Referee #2 · 28 Nov 2017

The subject of the ms is a revision of the shipboard splice of ODP Site 982 and some of its direct implications. Site 982 represents one of the most important sites, if not the most important one, to study paleoclimate change for the interval between 8 and 5 Ma in the critical North Atlantic. The interval notably covers the entire Messinian stage and its salinity crisis. Although the title may sound boring for some, this paper is critically important as it highlights the current tendency to revise shipboard splices, using high-resolution land-based core scanning data that are more suitable for splicing than the initial shipboard generated data. This tendency has major consequences for the paleoclimate and -oceanographic and IODP drilling community for instance regarding sample party and strategy, astronomical age models, etc. As such the paper should

serve as an eye opener for the community. However, it also shows the time consuming work that is behind the revision of such a shipboard splice, work that does not always seem to be valued. But, in this case, the implications of the revised splice discussed in the ms already make it perfectly clear why such a revision of the shipboard splice should become standard in the procedures of deep-sea drilling legs dedicated especially to paleoclimatic and -oceanographic studies.

The ms itself is clearly written and easy to follow. I only have one major issue as well as some minor ones. The major issue deals with the presentation of the tuning used to establish the astronomical age model. Following an initial age model based on calcareous plankton events, a minimal tuning is presented with approximately one tie-point per 100-kyr. This strategy is used to avoid incorporation of the amplitude modulation of precession by eccentricity in the tuned time series. In the first place, it might be added that the ages of the bio-events represent astronomically calibrated ages, which will facilitate tuning if these ages are (near) correct. The selected tie-points are shown in green in Figure 5 together with additional tie-points (in red) that were subsequently added to generate a next higher resolution astronomical age model. However, it is not made clear how and why the tie-points were selected and this should be made clear in the ms. In other words, what were the criteria and the approach used to select the tie-points for the tuning. The strategy of avoiding the amplitude modulation of precession to enter the tuned time series may suggest that the expression of the short eccentricity cycle itself might have played a central role in the selection of the tieÂň-points every ∼100-kyr. However, the expression of the short eccentricity itself is only present in part of the studied interval and thus seems not to have been used in the procedure, at least not over the entire interval. In addition, a minimal tuning might not be necessary as the modulation effect can be avoided by applying appropriate (wide) filters. Finally, it should be realized that the availability of astronomical ages for the bio-events as well as previously published age models may have played an important role in constraining the initial tuning and selecting the tiepoints.

Minor issues.

1) Some data are not fully shown in the figures in the Supplementary Information as they fall somehow outside the range of plotted values;

2) Add minor ticks on some of the x-axis in the Supplementary Figures, especially figure 3;

3) It might be preferable to use an offset between the isotope records in Figures 8A-B, and;

4) Make sure that there is a space between the genus and species name, also when the first one is abbreviated (i.e. in 2.4).

---

## Author Comment (AC2) · 2 Dec 2017

**Dear reviewer,**

**Thank you very much for your constructive review, which will help to improve our manuscript. Our replies to your comments (*in italics*) are shown below in red.**

*The subject of the ms is a revision of the shipboard splice of ODP Site 982 and some of its direct implications. Site 982 represents one of the most important sites, if not the most important one, to study paleoclimate change for the interval between 8 and 5 Ma in the critical North Atlantic. The interval notably covers the entire Messinian stage and its salinity crisis. Although the title may sound boring for some, this paper is critically important as it highlights the current tendency to revise shipboard splices, using high-resolution land-based core scanning data that are more suitable for splicing than the initial shipboard generated data. This tendency has major consequences for the paleoclimate and -oceanographic and IODP drilling community for instance regarding sample party and strategy, astronomical age models, etc. As such the paper should serve as an eye opener for the community. However, it also shows the time consuming work that is behind the revision of such a shipboard splice, work that does not always seem to be valued. But, in this case, the implications of the revised splice discussed in the ms already make it perfectly clear why such a revision of the shipboard splice should become standard in the procedures of deep-sea drilling legs dedicated especially to paleoclimatic and -oceanographic studies.*

*The ms itself is clearly written and easy to follow. I only have one major issue as well as some minor ones. The major issue deals with the presentation of the tuning used to establish the astronomical age model. Following an initial age model based on calcareous plankton events, a minimal tuning is presented with approximately one tie-point per 100-kyr. This strategy is used to avoid incorporation of the amplitude modulation of precession by eccentricity in the tuned time series. In the first place, it might be added that the ages of the bio-events represent astronomically calibrated ages, which will facilitate tuning if these ages are (near) correct. The selected tie- points are shown in green in Figure 5 together with additional tie-points (in red) that were subsequently added to generate a next higher resolution astronomical age model. However, it is not made clear how and why the tie-points were selected and this should be made clear in the ms. In other words, what were the criteria and the approach used to select the tie-points for the tuning. The strategy of avoiding the amplitude modulation of precession to enter the tuned time series may suggest that the expression of the short eccentricity cycle itself might have played a central role in the selection of the tie-points every ~100-kyr. However, the expression of the short eccentricity itself is only present in part of the studied interval and thus seems not to have been used in the procedure, at least not over the entire interval. In addition, a minimal tuning might not be necessary as the modulation effect can be avoided by applying appropriate (wide) filters. Finally, it should be realized that the availability of astronomical ages for the bio-events as well as previously published age models may have played an important role in constraining the initial tuning and selecting the tie-points.*

We appreciate that a number of things require clarifying in the astrochronology section. The first-order age model, generated using a polynomial fit through the shipboard nannofossil and planktonic foraminiferal datums (updated to the astronomical ages from Hilgen et al., 2012) was solely used to establish whether the strong ~0.8, ~1.6 and the 3.8-5.0 m cycles observed in the $\delta^{18}$O and $\delta^{13}$C records were most likely associated with astronomical forcing. We will clarify this in the text.

We did not use the polynomial age model itself as a starting point for the tuning. As we described in the text, we directly tuned the new benthic $\delta^{18}$O record from 982 to a E+T-P tuning target, specifically correlating $\delta^{18}$O minima to E+T-P maxima. As only the original dataset was used and no filters of the dataset were used, we did not feel additional explanation was required. However, we will adapt the text to provide a more thorough description of the tuning process, which we describe below in further detail.

The correlation of benthic $\delta^{18}O$ minima to ETP maxima was done visually, going directly from depth to age, facilitated by the tuning functions contained within CODD (Wilkens et al., 2017). The shipboard datums (Supplementary Table 6) were used to guide the correlation, however, these datums were not used as definitive tie points. As the shipboard datums have considerable depth errors (between ±0.25-2.5 m; we will adjust Supplementary Table 6 to include these errors), we did not feel these datums were reliable enough to use as rigid guides for tie point allocation. As such, we also considered the influence of the astronomical calibration of these biostratigraphic datums on our tuning to be minimal, although the reviewer is correct that this influence cannot entirely be excluded.

For the initial tuning, we visually correlated distinctive $\delta^{18}O$ cycles to ETP maxima with a correspondingly distinctive shape, resulting from the interference patterns between obliquity and precession. The minimal-tuning tie points were chosen to align all $\delta^{18}O$ minima and ETP maxima as best as possible across the entire record, especially between the tie points. We tried to use as few tie points as possible for the minimal tuning, following the strategy outlined in Holbourn et al., 2007. We thereby also tried to leave at least ~100 kyr between consecutive minimal tuning tie points, as not to introduce frequency modulation into the record, as outlined in Zeeden et al., 2015. We will particularly rephrase and clarify the part of the text relating to this, as we did not choose a minimal tuning tie approximately ~100 kyr, but rather made sure consecutive minimal tuning ties were ideally at least 100 kyr apart (actual range of time between consecutive ties = 90-377 kyr). As such, we do not believe that short-term eccentricity was important in the selection of our ties. However, we will make sure this misunderstanding is clarified in the text.

The same strategy in visually correlating the $\delta^{18}O$ minima to ETP maxima was employed to obtain the fine-tuning age model, thereby providing higher-resolution age control and remove any remaining misalignments between $\delta^{18}O$ cycles and the ETP curve in between the minimal tuning tie points. We will adapt the current text to clarify this further.

Although there are other ways to avoid amplitude modulation, as the reviewer suggests, we chose to provide the complimentary minimal and fine-tuning age models and allow the reader to choose the tuning strategy that best fits their application (e.g. minimal tuned age model for reconstructing changes in phase over time, versus the fine-tuned age model for high-resolution correlation between different records).

*Minor issues.*
*1) Some data are not fully shown in the figures in the Supplementary Information as they fall somehow outside the range of plotted values;*
We will adjust the y-axes of the $\delta^{13}C$ in Panels 6 and 7 of Supplementary Figure 2.

*2) Add minor ticks on some of the x-axis in the Supplementary Figures, especially figure 3;*
Will add minor ticks to all three supplementary figures. We will additionally revisit the figures in the main manuscript to improve this where necessary.

*3) It might be preferable to use an offset between the isotope records in Figures 8A-B, and;*
We prefer not to add an offset, as we feel that the overlap in the data shows the disagreement better in panel A and shows the agreement better in panel B.

*4) Make sure that there is a space between the genus and species name, also when the first one is abbreviated (i.e. in 2.4).*
Thanks for pointing this out. We were not consistent in our use of a space when referencing the benthic foraminiferal species. We will rectify this throughout the text. We additionally defined that *C.* stands for *Cibicidoides*.

---

## Author Response (AR1)

**Point by point overview of minor revisions to**
**"Reinforcing the North Atlantic backbone: revision and extension of the composite splice at ODP Site 982"**

We have completed the minor revisions to our manuscript, as outlined in our author comments in response to the two reviewer comments during the interactive discussion. We have summarised the changes here.

*In response to Reviewer 1:*

Main MS, Pg 1 – Ln 24-26: we included that the scheme for late Miocene marine isotope stages needs redefining. This was also incorporated in the conclusions (Pg 10 – Ln 24-25).

Main MS, Pg 3 – Ln 14: we have replaced "proximity to the Mediterranean Sea" with "location in the North Atlantic".

Main MS, Pg 4 – Ln 8-11: as requested, we have extended this section by integrating information originally included on Pg 5 – Ln 12 to clarify the strategy employed in generating a new, complete benthic stable isotope stratigraphy at Site 982.

Main MS, Pg 5 – Ln 2, 5-6: we have now included the revised offset and splice tables in the main manuscript, as requested. We have additionally provided digital .xlsx files in the supplement, as these will be more convenient for later use.

Main MS, Pg 5 – Ln 15, Figure 5 (previously 7) and throughout: we have now included MTM spectra of the d18O and d13C series in the depth domain in Figure 5 (Previously Figure 7). This has resulted in a reordering of the figure numbers relative to the Discussion manuscript.

Main MS, Pg 7 – Ln 6, 9-10, 25 and Figure 7 (previously 6): we have included a coherency wavelet of the d18O-d13C data from Site U1337 to illustrate the antiphase obliquity relationship present throughout the record.

Main MS, Pg 9 – Ln 20-21: we have extended this sentence to clarify that lower sampling resolution and lower sedimentation rates at Ain El Beida (AEB) are more likely the cause of the discrepancy in the number of cycles seen at AEB vs Site 982 between TG12/14 and TG20/22.

Main MS, Pg 10 – Ln 3-27: we have shortened the conclusions.

Figure 2: we have now included the GRA bulk density data collected during the expedition to illustrate the shipboard splice choices, as requested by the reviewer. Unfortunately, it was not possible to include the reflectance data, as we were not able to access uncorrupted data, even through the ODP database librarian. However, we feel that the shipboard GRA data successfully illustrates why certain shipboard choices were made, but that these require revision based on new data that highlights offsets that are not clearly visible in the shipboard data.

Figure 4: we have included the missing "E" and capitalised all subfigure labels in the figures and the figure captions.

Figure 6 (Previously 5) and Supplementary Figure 4 (Previously SF 3): we have moved the minimal and fine tuning labels from the bottom right to the top left, so they are more prominent. We have additionally changed the tie point colours from green/orange to purple/grey to make the tiepoints easier to distinguish. We have finally added "benthic" to every applicable y-axis (also in other figures).

Figure 7 (Previously 6): we have shifted the red arrows to the correct position and added an age label to the x-axis for Panel A.

***In response to Reviewer 2:***
Main MS, Pg 4 – Ln 12: we have defined that *C.* stands for *Cibicidoides*, and included a space after each mention of *C.*.

Main MS, Pg 6 – Ln 1-28: as described in greater detail in the Author comment in response to Reviewer 2's comment, we have considerably rewritten the Astrochronology section. We have included the additional information requested by the reviewer and sought to clarify the approach we used to obtain minimal and fine tuned astrochronologies.

Figures 3, 4, 6, 7, 8, Supplementary Figures 2, 3, 4: we have now included additional minor ticks along the x-axis in all these figures.

Supplementary Figure 3 (previously SF 2): we have adjusted the y-axis of the d13C data so all data now appears without being cut off.

***Additional changes:***
Minor changes (i.e., figure number changes etc.) have been made throughout the manuscript (see the track changes document).

Main MS, Pg 3 – Ln 1-2: we have now included the DOI information for the dataset on PANGAEA. Also the case for Pg 11 – Ln 2-3 and 18-19.

Main MS, Pg 3 – Ln 27: we have now included a reference to the supplementary information, where we describe the XRF calibration in more detail.

Supplementary information: we have included a new section describing the XRF calibration and providing the calibration formulas. Additionally, we have provided the code used to generate the MTM spectra in Astrochron.

[revised manuscript text omitted]

a

| Page 25: [2] Deleted | Anna Joy Drury | 28/11/2017 16:39:00 |

a

| Page 25: [2] Deleted | Anna Joy Drury | 28/11/2017 16:39:00 |

a

| Page 25: [2] Deleted | Anna Joy Drury | 28/11/2017 16:39:00 |

a

| Page 25: [2] Deleted | Anna Joy Drury | 28/11/2017 16:39:00 |

a

| Page 25: [2] Deleted | Anna Joy Drury | 28/11/2017 16:39:00 |

a

---

## Author Response (AR2)

**Dr Anna Joy Drury**
*Postdoctoral Researcher*
*Phone:* +49 (0) 421 218 65785
*Email:* ajdrury@marum.de

MARUM II, Room 0020
Leobener Strasse
28359 Bremen
Germany

Bremen, 17.01.2018

**Subject:** Resubmission of minor revisions to *Climate of the Past*

Dear Pierre,

We hereby submit the revision for our manuscript entitled '*Reinforcing the North Atlantic backbone: revision and extension of the composite splice at ODP Site 982*' for publication in *Climates of the Past.* An additional supplementary information file is provided, including supplementary figures 1-4 and tables 2 and 6. All data produced in this manuscript is also provided in an excel file (supplementary tables 1-7). These tables, together with a full Site 982 CODD experiment and the composite core photos as IGOR binary files and PDFs will be available from the open access PANGAEA database at DOI https://doi.org/10.1594/PANGAEA.884300. This DOI will become active once the accompanying manuscript is officially published. Until that time, the data is available at https://doi.pangaea.de/10.1594/PANGAEA.884300, although the datasets themselves remain password protected.

Thank you very much for your constructive feedback in the last review. The comments and suggestions were helpful and helped to improve the manuscript further (please see our detailed response below). We are additionally very happy to hear that the content of the manuscript was well-received. We hope that the manuscript in its present form provides a well-illustrated example for researchers utilising high-resolution palaeoceanographical records that it is crucial to verify shipboard splices from legacy ocean drilling sites to ensure stratigraphic completeness of their records. We additionally hope that the new stratigraphies presented here prove useful for future palaeoceanographic research in the North Atlantic.

All authors agree to manuscript submission. Thank you again for your helpful review.

I look forward to hearing from you.

Many thanks and kind regards,

Anna Joy Drury
*Corresponding author, MARUM, University of Bremen*

**Point by point overview of minor revisions to "Reinforcing the North Atlantic backbone: revision and extension of the composite splice at ODP Site 982"**

We thank Pierre for his helpful suggestions. We have implemented almost all requested changes, or explained otherwise in our more detailed response (in blue) to each comment (in black) below.

***Response to editor comments in online review:***
One thing that needs to be addressed is the reference style, that is not following the format required by Clim. Past.
Please also see our response below, we have corrected this.

I noted that I cannot access the data in Pangaea following the reference given. Maybe it is because you have not made the data available yet, or maybe the ling/doi ref. is wrong. Please check this.
The DOI is not currently active, as it requires the dataset to be fully published before the DOI is activated. The unpublished dataset is currently available (password protected I think) at this temporary DOI: https://doi.pangaea.de/10.1594/PANGAEA.884300. We did not include this in the revised submission as it will become defunct once the dataset is published and the DOI in the manuscript becomes active. The DOI link in the manuscript will provide an open access link to the dataset once the accompanying paper is published.

Finally, the Figure/Table numbering is not identical from one figure to another, including in the supplement. Please try having the same labelling, in the caption and in the figure itself (you often mix "a" and "A").
Please also see our response below, we have corrected this.

***Response to editor comments on PDF:***

Pg 1 – Ln 13: insert "that was filled with in with 263 new isotope analyses" and replace "Our" with "This"
We have made the suggested revisions, and additionally added "benthic foraminiferal" to line 11 to clarify that we are generating a benthic foraminiferal stable isotope stratigraphy (as opposed to a planktic foraminiferal or bulk stable isotope stratigraphy).

Pg 3 – Lns 2-4: I suggest to move this information to the Date availability section to avoid unnecessary repetition.
We have deleted this information here, and additionally added the reference to the JANUS database in the Data availability section.

Pg 3 – Lns 15-16: Why do you want to compare with these records. Because they are the nearest available records of the same resolution. If it is the case, please write it. (although I acknowledge you explain it in the discussion section).
Thank you for this suggestion. We have firstly elaborated that we assess the new Site 982 stratigraphy using the stable isotope stratigraphies from ODP Site 926 and Ain el Beida, which indeed are the nearest records with comparable resolution and robust astrochronologies. Additionally, we have included a sentence to explain that these records have additionally been instrumental in initially highlighting that there were stratigraphic issues with the shipboard splice at Site 982.

Pg 4 – Ln 3: Please add or delete a parenthesis
We have made the suggested revisions.

Pg 4 – Ln 4: Please add the link
We have made the suggested revisions.

Pg 4 – Ln 25: These photos are barely visible. I understand that you do not want to make them too visible to obliterate the plots, but is it possible to do something better? At least explain why they look black & white (are they originally or did you transformed them in B&W, and if you did, please explain how)
The composite core photos are the original colour of the sediment, which are in full colour. Due to the high carbonate content, the original sediment colour is simply that light and hard to resolve against any background. Nothing was done to artificially lighten the sediments, and they appear in the figures and supplement/PANGAEA in their original colour. We have added a sentence at this location to report that the low ln(Si) values coincide with the lighter, whiter layers.
The CODD functions for Core_Table_Photos includes a lighting correction to solely removes artificial shadows due to the use of a single point lighting source on ODP core table top photos, which would otherwise imprint a ~1.5 m artificial cyclicity into the composite core photos. We have added a reference to the Wilkens et al., 2017 in Section 2.3 to indicate where the reader can find further information on the Lighting Correction function.

Pg 4 – Ln 26: In the figures you use A, B, C,... Might be good to do the same in the text instead of using lowercase letters
We have made the suggested revisions.

Pg 5 – Ln 8: Please add "; Figure 5A"
We have made the suggested revisions.

Pg 5 – Ln 8: This is barely seen in Figure 5. I would not qualify this as "Strong"
We have removed both uses of "strong" in this sentence, and replaced the first mention with "distinct" and the second mention with "visible". We

Pg 6 – Ln 32, Pg 7 – Ln 3 and 4: Delete "."
We have made the suggested revisions.

Pg 11- Lns 16-18: Again please put that information only in the data availability section.
We have removed this information. It is currently only referenced in the data availability section. We have additionally extended the acknowledgements to include the reviewers and editor for their constructive feedback.

References: Years of publication should be at the end of the reference (I know, this is strange). Please format the references according to the instructions. Journal titles should be abbreviated according to ISI Journal title abbreviation. Also see this document: https://www.climate-of-the-past.net/Copernicus_Publications_Reference_Types.pdf
Thank you for pointing this out, we have put the references in the correct format.

Pg 20 – Ln 8: There is no image in there, only plots.
We used "image" here as interchangeable with "figure", not specifically to indicate the inclusion of composite core images. We have however replaced "images" with "plots".

Pg 22 – Figure 5: The depth and the time interval on which these analyses have been performed is not specified. I presume it is on the whole interval, but please indicated it here in the caption.
We have noted in the caption that the spectra are of of the entire stable isotope stratigraphies (8.0-4.5 Ma).

Pg 24 – Ln 7: please correct the reference style for Grinsted et al., 2004.
We have made the suggested revisions.

[revised manuscript text omitted]